# KK-LC-1 as a therapeutic target to eliminate ALDH⁺ stem cells in triple negative breast cancer

Jiawen Bu[1,6], Yixiao Zhang[1,6], Sijin Wu[1,2,6], Haonan Li[3,6], Lisha Sun[1,6], Yang Liu[4,5], Xudong Zhu[1], Xinbo Qiao[1], Qingtian Ma[1], Chao Liu[1], Nan Niu [1], Jinqi Xue[1], Guanglei Chen[1], Yongliang Yang [1,3] ✉ & Caigang Liu [1] ✉

Failure to achieve complete elimination of triple negative breast cancer (TNBC) stem cells after adjuvant therapy is associated with poor outcomes. Aldehyde dehydrogenase 1 (ALDH1) is a marker of breast cancer stem cells (BCSCs), and its enzymatic activity regulates tumor stemness. Identifying upstream targets to control ALDH⁺ cells may facilitate TNBC tumor suppression. Here, we show that KK-LC-1 determines the stemness of TNBC ALDH⁺ cells via binding with FAT1 and subsequently promoting its ubiquitination and degradation. This compromises the Hippo pathway and leads to nuclear translocation of YAP1 and ALDH1A1 transcription. These findings identify the KK-LC-1-FAT1-Hippo-ALDH1A1 pathway in TNBC ALDH⁺ cells as a therapeutic target. To reverse the malignancy due to KK-LC-1 expression, we employ a computational approach and discover Z839878730 (Z8) as an small-molecule inhibitor which may disrupt KK-LC-1 and FAT1 binding. We demonstrate that Z8 suppresses TNBC tumor growth via a mechanism that reactivates the Hippo pathway and decreases TNBC ALDH⁺ cell stemness and viability.

Triple-negative breast cancer (TNBC) is a highly heterogeneous and clinically aggressive disease that accounts for 15–20% of all breast cancers[1,2]. Conventional chemotherapy remains as the standard of care. Unfortunately, responses to conventional treatment are short with frequent visceral and brain metastatic spread[3]. There remains an urgent yet unmet medical need to identify an effective targeted therapy for TNBC, especially cancer cells which could escape adjuvant therapy.

Cancer cells vary in their susceptibility to chemotherapy. On the other hand, chemotherapy resistance can lead to multi-drug resistance, cancer recurrence, and poor clinical outcomes[4–6]. Cancer stem cells (CSCs) are more resistant to chemotherapy than other cancer cells and played a crucial role in breast cancer progression, drug resistance, and relapse[7]. Hence, therapies that specifically target BCSC populations may reduce breast cancer drug resistance and metastasis[8]. BCSCs consist of small populations of CD44^high CD24^low cells and cells with high aldehyde dehydrogenase 1 (ALDH1) activity[9,10]. The plasticity of BCSCs allows them to transition between the two CSC states[11]. ALDH⁺ cells are more prone to create colonies with greater tumorigenic potential and are more drug-resistant than CD44^high CD24^low cells[9,12]; however, the underlying mechanisms that regulate ALDH⁺ cells and the transition between ALDH⁺ cells and CD44^high CD24^low cells remain to be elucidated. Hence, targeted therapeutics that can effectively eradicate ALDH⁺ cells with minimal side effects are urgently needed for TNBC treatment.

[1]Cancer Stem Cell and Translation Medicine Lab, Department of Oncology, Innovative Cancer Drug Research and Development Engineering Center of Liaoning Province, Shengjing Hospital of China Medical University, 110004 Shenyang, China. [2]Shenzhen Jingtai Technology Co., Ltd. (XtalPi), International Biomedical Industrial Park (Phase II) 3F, 2 Hongliu Rd, Futian District, 16023 Shenzhen, China. [3]School of Bioengineering, Dalian University of Technology, 116023 Dalian, China. [4]School of Pharmaceutical Engineering, Shenyang Pharmaceutical University, 110016 Shenyang, China. [5]Key Laboratory of Structure-Based Drug Design and Discovery of Ministry of Education, Shenyang Pharmaceutical University, 110016 Shenyang, China. [6]These authors contributed equally: Jiawen Bu, Yixiao Zhang, Sijin Wu, Haonan Li, Lisha Sun. ✉e-mail: everbright99@foxmail.com; liucg@sj-hospital.org

Kita-Kyushu lung cancer antigen-1 (KK-LC-1), also known as CT83 or Cxorf61, is a cancer-testis antigen (CTA)[13]. Emerging evidence has provided a theoretical basis for the development of therapies targeting KK-LC-1 in various tumors[14]. KK-LC-1 is abnormally expressed in lung cancer, liver cancer, gastric cancer, and breast cancer. KK-LC-1 has been identified as a prognostic biomarker in several solid tumors[14,15]. In hepatocellular carcinoma (HCC), high KK-LC-1 expression levels are associated with cell growth, invasion, migration, epithelial–mesenchymal transition (EMT), and poor survival outcomes[16]. In gastric cancer, high levels of KK-LC-1 protein expression are associated with longer overall survival[14]. KK-LC-1 is expressed in 65% of TNBC. The mechanism of action of KK-LC-1 in TNBC has yet to be defined[17].

In this work, we show that KK-LC-1 dictates the stemness of TNBC ALDH⁺ cells and serves as a promising therapeutic target in female TNBC. We find that KK-LC-1 is highly expressed in ALDH⁺ cells from TNBC. After *KK-LC-1* knockout, the proportion of ALDH⁺ cells is significantly decreased, the ability of CD44^high CD24^low cells to transition to ALDH⁺ cells is reduced, and ALDH⁺ cells have significantly impaired self-renewal and sphere-forming capacity. Mechanistically, KK-LC-1 is able to bind with FAT1 which leads to FAT1 ubiquitination and subsequent degradation. FAT1 is an upstream regulator of the Hippo pathway[18]; therefore, degradation of FAT1 compromises the Hippo pathway, and YAP1 is translocated to the nucleus to increase transcription. The stemness transcription factor SOX2 is significantly upregulated and ALDH1A1 transcript level increases, resulting in a higher proportion of ALDH⁺ cells and TNBC cells with aggressive features. Notably, we identify a small-molecule compound as a protein–protein interface inhibitor (PPI) for KK-LC-1/FAT1 complex via a computational approach. We demonstrate that Z8 may disrupt the binding of KK-LC-1 and FAT1, which in turn protects FAT1 from degradation and maintains the Hippo pathway in an active state. When the Hippo pathway is activated, ALDH1A1 expression is inhibited, and the proportion of ALDH⁺ cells is decreased.

## Results

### Cancer testis antigen KK-LC-1 is enriched in TNBC ALDH⁺ cells

Drug resistance has become an intractable obstacle in the treatment of TNBC and lacking of specific targets to drug-resistant TNBC cells (Supplementary Fig. 1a) made them hard to eliminate. To seek potential targets to eliminate TNBC cells that could survive chemotherapy, docetaxel-resistant MDA-MB-231 cells were subjected to mRNA-seq along with their parental cell line (Fig. 1a). Among the up-regulated genes in docetaxel-resistant MDA-MB-231 and up-regulated genes in TNBC (Supplementary Data 1) compared to normal tissue[19,20], we identified two overlapped high up-regulated genes, *MMP1* and *KK-LC-1* (Fig. 1b, c). As cells vary in their susceptibility to chemotherapy and cancer stem cells especially contribute to drug resistance, we hypothesized if the overlapped up-regulated genes in docetaxel-resistant MDA-MB-231 as well as in TNBC tissue may be related to cancer stem cells in breast cancer. Consequently, we examined the expression of *MMP1* and *KK-LC-1* in CD44^high CD24^low cells or ALDH⁺ cells in TNBC PDX (patient-derived xenografts) models. In brief, TNBC PDX tumors were harvested, digested into single cell suspensions, and then separated into ALDH⁺ and ALDH⁻ groups or CD44^high CD24^low and non-CD44^high CD24^low group by FACS sorting (Fig. 1d, Supplementary Fig. 1b). To investigate the putative association of *KK-LC-1* and *MMP1* with cancer stemness, we examined the expression of *KK-LC-1* and *MMP1* in various sorted population of cells. Noteworthy, *KK-LC-1* was found specifically overexpressed in the purified ALDH⁺ population with 10,000 fold higher than ALDH⁻ population (Fig. 1e), indicating that *KK-LC-1* may plays an important role in the regulation of ALDH⁺ cells, a subpopulation of cells enriched in BCSCs. In contrast, we did not find a significant

difference in expression for *MMP1* in various populations of cells (Fig. 1e and Supplementary Fig. 1c). Analysis of the TCGA database showed that *KK-LC-1* was overexpressed in the tumors than its normal counterparts (Supplementary Fig. 1d)[19,20]. Consistently, immunoblotting confirmed higher KK-LC-1 protein expression in the TNBC tumor tissues than in the para-cancerous tissues (Supplementary Fig. 1g). Breast cancer patients with higher *KK-LC-1* mRNA expression had poorer overall survival (Supplementary Fig. 1e)[21] and KK-LC-1 expression was also associated with worse disease-free survival as well as overall survival in TNBC patients (Supplementary Fig. 1f). Moreover, our western results confirmed higher KK-LC-1 protein expression in the TNBC tumor tissues than in the para-cancerous tissues (Supplementary Fig. 1g). Furthermore, KK-LC-1 protein was found overexpressed specifically in the TNBC cell lines MDA-MB-231 and MDA-MB-468 compared to multiple non-TNBC breast cancer cell lines, including the ER⁻HER2⁺ breast cancer cell line SKBR3, the ER+ breast cancer cell line MCF7, the ER⁺HER2⁺ breast cancer cell line BT474 and the breast epithelial cell line MCF10A (Supplementary Fig. 1h, i)[22]. Collectively, these data identified KK-LC-1 as an important marker for ALDH⁺ cells and more specifically for the BCSCs of TNBC.

### KK-LC-1 functionally regulates the stemness of TNBC ALDH⁺ cells

To determine the functional role of KK-LC-1 in the ALDH⁺ cells, *KK-LC-1* knockout was performed in MDA-MB-231 and MDA-MB-468 cells via CRISPR/Cas9 (Supplementary Fig. 1j). The proportions of ALDH⁺ cells were significantly reduced in *KK-LC-1*⁻/⁻ MDA-MB-231 and MDA-MB-468 cells compared to their parental cell line (Fig. 1f, g, Supplementary Figs. 2 and 3a, b). The proportion of CD44^high CD24^low population in the ALDH⁺ cells was also significantly decreased with *KK-LC-1* knockout (Fig. 1f, g and Supplementary Fig. 3a, b). We next investigated whether KK-LC-1 can affect the self-renewal properties of ALDH⁺ cells in TNBC. ALDH⁺ cells isolated from MDA-MB-231 cells were transduced with sh-*KK-LC-1*#1 lentivirus or negative control (NC) lentivirus (knockdown efficacy, approximately 80%) (Supplementary Fig. 3c, d), followed by in vitro culture for 2 weeks. Flow cytometry and sphere formation under limiting dilution conditions showed that *KK-LC-1* knockdown significantly reduced the self-renewal capacity of ALDH⁺ cells at 2 weeks (Fig. 1h and Supplementary Table 1). Strikingly, the knockdown of *KK-LC-1* diminished the drug resistance of docetaxel-resistant MDA-MB-231 cells to docetaxel (Supplementary Fig. 3e) and decreased the proportion of ALDH⁺ cells (Supplementary Fig. 3f, g). Because ALDH⁺ cells and CD44^high CD24^low cells can transit to each other, we tested whether the knockout of *KK-LC-1* on CD44^high CD24^low cells affects its transition to ALDH⁺ cells. Knockout of *KK-LC-1* weakened the ability of CD44^high CD24^low cells to transit to ALDH⁺ cells in MDA-MB-231 cells (Fig. 1i). The ratio of CD44^high CD24^low cells preserved in the sorted-out cells was also higher in the *KK-LC-1*⁻/⁻ MDA-MB-231 group compared to *KK-LC-1*⁺/⁺ MDA-MB-231 group, yet not statistically significant (Supplementary Fig. 3h, i).

Sphere formation has been associated with the expression of stemness-related genes and CSC markers in multiple tumors[23]. We performed sphere formation assays to explore if KK-LC-1 maintains the stemness of ALDH⁺ cells in TNBC. Purified ALDH⁺ cells from MDA-MB-231 and MDA-MB-468 cell lines were transduced with sh-*KK-LC-1*#1, sh-*KK-LC-1*#2, and negative control (NC) lentivirus. ALDH⁻ MDA-MB-231 and MDA-MB-468 cells were also treated with the same intervention and served as negative controls. The numbers and diameters of spheres were significantly reduced in sh-*KK-LC-1*#1 and sh-*KK-LC-1*#2 ALDH⁺ cells compared to the counterpart transduced with negative control (NC) lentivirus (Fig. 2a–d). In contrast, ALDH⁻ cells could barely form tumor spheres and the group transduced with sh-*KK-LC-1*#1 and sh-*KK-LC-1*#2 exhibit similarly sphere formation ability as compared to the negative control (NC) group (Fig. 2a–d). ALDH1A1 is an isoenzyme

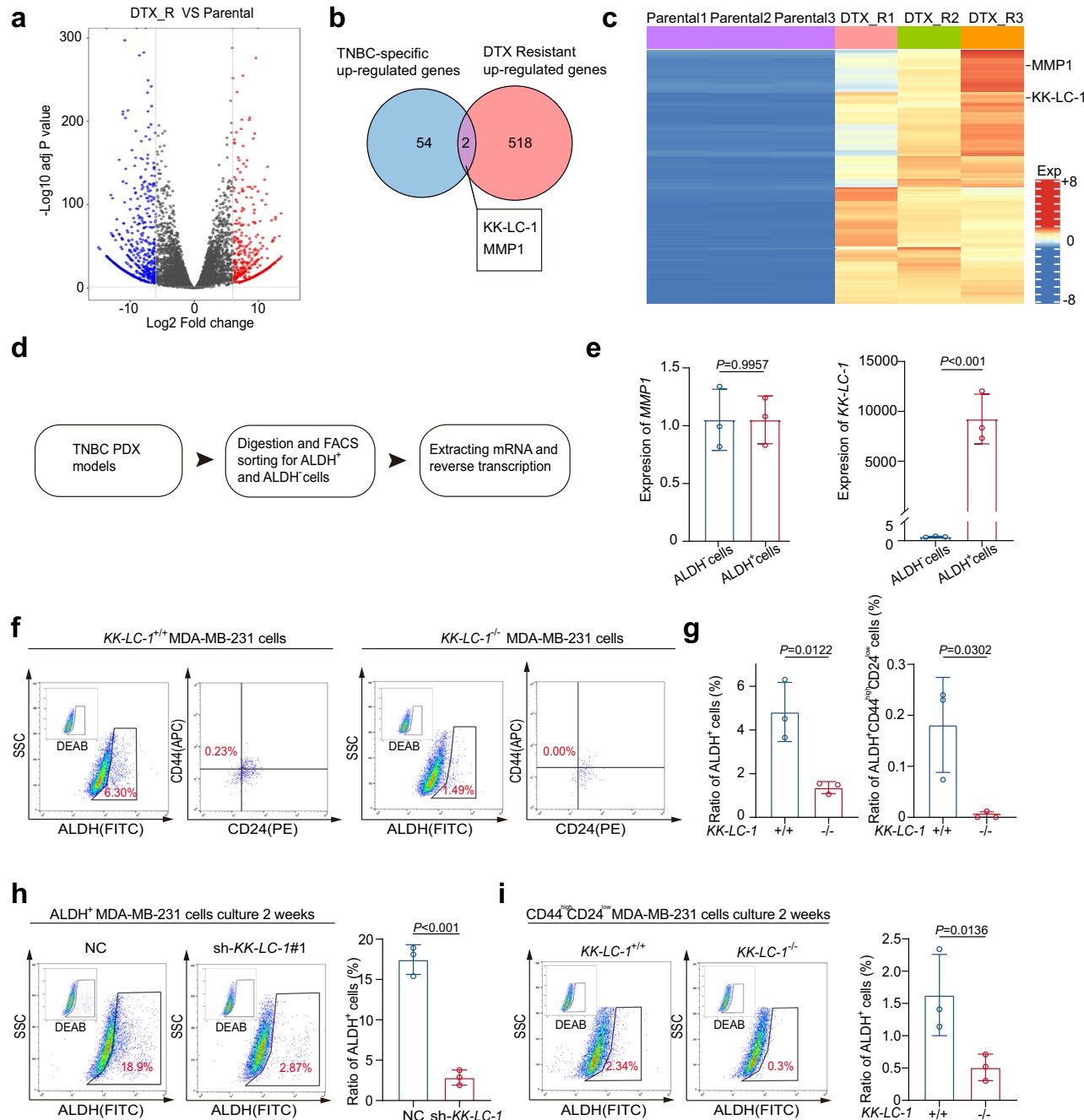

**Fig. 1 | KK-LC-1 regulated the proportion, self-renewal, and generation of ALDH⁺ cells in TNBC. a** Volcano plot showing up-regulated genes in docetaxel-resistant MDA-MB-231 cells (DTX_R) compared to their parental cells (Parental) (Log2 Fold change > 6, adj *P* value < 0.05, *P* values were determined using moderated *t*-statistic, adj *P*-value was calculated using Benjamini and Hochberg's method and tails were two-sided). **b** Up-regulated genes in docetaxel-resistant MDA-MB-231 cells were intersected with the up-regulated genes in TNBC[19,20]. Expression levels of two genes were significantly higher in docetaxel-resistant MDA-MB-231 cells. **c** Heatmap showing expression of the two highly expressed genes. **d** Schematic showing the identification and isolation of primary ALDH⁺ and ALDH⁻ cells. **e** Among the two highly expressed genes, *KK-LC-1* had the highest expression in ALDH⁺ cells compared to ALDH⁻ cells and was selected as a potential candidate that may regulate ALDH⁺ cells in TNBC; data are presented as mean ± SD from three biologically independent experiments, statistical significance was determined using Student's

*t*-test and was two-sided. **f, g**: *KK-LC-1* knockout significantly reduced the proportion of ALDH⁺ cells and ALDH⁺CD44highCD24low cells in MDA-MB-231 cells; data are presented as mean ± SD from three biologically independent experiments and statistical significance was determined using Student's *t*-test and was two-sided. **h** The proportion of ALDH⁺ cells was significantly lower in ALDH⁺ sh-*KK-LC-1*#1 cells compared to ALDH⁺ NC cells after 2 weeks in culture; data are presented as mean ± SD from three biologically independent experiments, statistical significance was determined using Student's *t*-test and was two-sided. **i** A significantly higher proportion of ALDH⁺ cells was derived from CD44highCD24low *KK-LC-1*⁺/⁺ MDA-MB-231 compared to CD44highCD24low *KK-LC-1*⁻/⁻ MDA-MB-231 cells after 2 weeks in culture; data are presented as mean ± SD from three biologically independent experiments, statistical significance was determined using Student's *t*-test and was two-sided. Source data are provided as a Source data file.

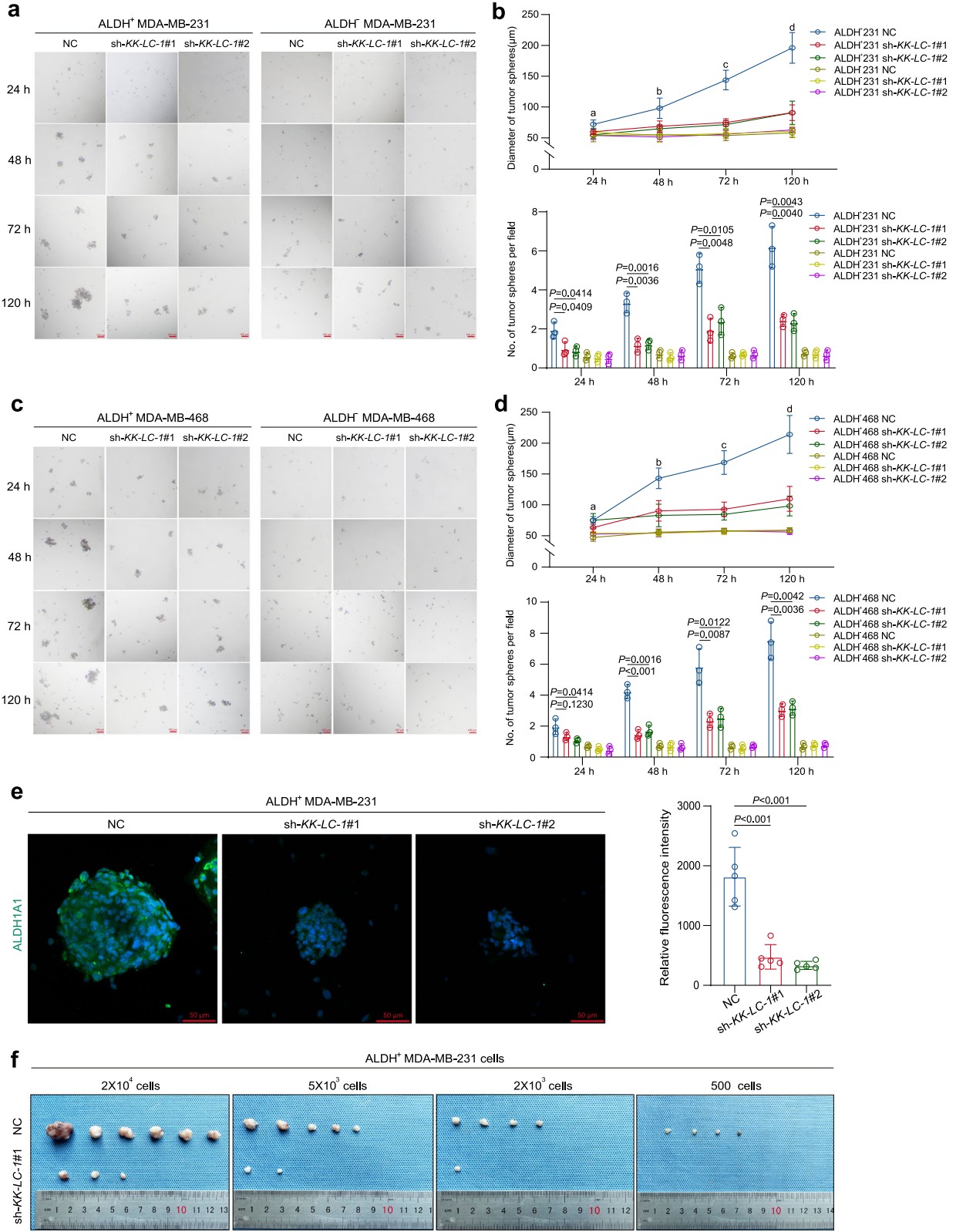

in the ALDH⁺ family that is associated with cancer ALDH⁺ stemness[24] and was found reduced in the sh-*KK-LC-1*#1 and sh-*KK-LC-1*#2 spheres compared to the control spheres (Fig. 2e). To determine the possible role of KK-LC-1 in the initiation and growth of BCSCs, xenograft transplantation assays were conducted under limiting dilution condition. ALDH⁺ MDA-MB-231 cells transduced with sh-*KK-LC-1*#1 had an approximately fifteen-fold decrease in the frequency of tumor-propagating cells compared to the cells transduced with control

(NC) lentivirus (Fig. 2e and Supplementary Table 2). Furthermore, *KK-LC-1* knockout significantly decreased the tumor growth in vivo, manifested by much smaller tumor volume and weight, compared to the control mice (Supplementary Fig. 3j–l). The levels of KK-LC-1 protein were dramatically suppressed in the *KK-LC-1*⁻/⁻ xenograft tumors, indicating that KK-LC-1 is required for BCSCs to initiate and propagate TNBC tumors (Supplementary Fig. 3m). Taken together, our results reveal that KK-LC-1 plays an important functional role to maintain the

**Fig. 2 | KK-LC-1 regulated TNBC ALDH+ cell tumorsphere formation capacity and tumorgenicity. a** Sphere formation of ALDH+ MDA-MB-231 cells and ALDH- MDA-MB-231 cells transduced with sh-*KK-LC-1*#1, sh-*KK-LC-1*#2, and negative control (NC) lentivirus at various time points, red scale bars indicate 100 μm. **b** Diameter and number of spheres formed by ALDH+ MDA-MB-231 cells and ALDH- MDA-MB-231 cells transduced with sh-*KK-LC-1*#1, sh-*KK-LC-1*#2 and negative control (NC) lentivirus at various time points; data are presented as mean ± SD from three biologically independent experiments. Regarding the diameter of the spheres, 'a' indicates *P* = 0.0712 when ALDH+ MDA-MB-231 sh-*KK-LC-1*#1 compared to ALDH+ MDA-MB-231 NC and *P* = 0.0415 when ALDH+ MDA-MB-231 sh-*KK-LC-1*#2 compared to ALDH+ MDA-MB-231 NC at 24 h time point, 'b' indicates *P* = 0.0525 when ALDH+ MDA-MB-231 sh-*KK-LC-1*#1 compared to ALDH+ MDA-MB-231 NC and *P* = 0.0297 when ALDH+ MDA-MB-231 sh-*KK-LC-1*#2 compared to ALDH+ MDA-MB-231 NC at 48 h time point, 'c' indicates *P* = 0.0021 when ALDH+ MDA-MB-231 sh-*KK-LC-1*#1 compared to ALDH+ MDA-MB-231 NC and *P* = 0.0019 when ALDH+ MDA-MB-231 sh-*KK-LC-1*#2 compared to ALDH+ MDA-MB-231 NC at 72 h time point, 'd' indicates *P* = 0.0028 when ALDH+ MDA-MB-231 sh-*KK-LC-1*#1 compared to ALDH+ MDA-MB-231 NC and *P* = 0.0043 when ALDH+ MDA-MB-231 sh-*KK-LC-1*#2 compared to ALDH+ MDA-MB-231 NC at 120 h time point, statistical significance was determined using Student's *t*-test and was two-sided. **c:** Sphere formation of ALDH+ MDA-MB-468 cells and ALDH- MDA-MB-468 cells transduced with sh-*KK-LC-1*#1, sh-*KK-LC-1*#2 and negative control (NC) lentivirus at various time points, red scale bars indicate 100 μm. **d:** Diameter and number of spheres formed by ALDH+ MDA-MB-468 cells and ALDH- MDA-MB-468 cells transduced with sh-*KK-LC-1*#1, sh-*KK-LC-1*#2 and

negative control (NC) lentivirus at various time points; data are presented as mean ± SD from three biologically independent experiments. Regarding to the diameter of the spheres, 'a' indicates *P* = 0.1089 when ALDH+ MDA-MB-468 sh-*KK-LC-1*#1 compared to ALDH+ MDA-MB-468 NC and *P* = 0.9846 when ALDH+ MDA-MB-468 sh-*KK-LC-1*#2 compared to ALDH+ MDA-MB-468 NC at 24 h time point, 'b' indicates *P* = 0.0178 when ALDH+ MDA-MB-468 sh-*KK-LC-1*#1 compared to ALDH+ MDA-MB-468 NC and *P* = 0.0137 when ALDH+ MDA-MB-468 sh-*KK-LC-1*#2 compared to ALDH+ MDA-MB-468 NC at 48 h time point, 'c' indicates *P* = 0.0043 when ALDH+ MDA-MB-468 sh-*KK-LC-1*#1 compared to ALDH+ MDA-MB-468 NC and *P* = 0.0023 when ALDH+ MDA-MB-468 sh-*KK-LC-1*#2 compared to ALDH+ MDA-MB-468 NC at 72 h time point, 'd' indicates *P* = 0.0008 when ALDH+ MDA-MB-468 sh-*KK-LC-1*#1 compared to ALDH+ MDA-MB-468 NC and *P* = 0.0044 when ALDH+ MDA-MB-468 sh-*KK-LC-1*#2 compared to ALDH+ MDA-MB-468 NC at 120 h time point, statistical significance was determined using Student's *t*-test and was two-sided. **e** Immunofluorescence staining showing the difference in ALDH1A1 protein expression in spheres formed by ALDH+ MDA-MB-231 cells transduced with sh-*KK-LC-1*#1, sh-*KK-LC-1*#2 and negative control (NC) lentivirus, red scale bars indicate 50 μm; data are presented as mean ± SD from five randomized views and this experiment was performed independently for 3 times, statistical significance was determined using Student's *t*-test and was two-sided. **f** Representative tumors formed by ALDH+ MDA-MB-231 cells transduced with sh-*KK-LC-1*#1 and negative control (NC) lentivirus in tumor propagation assay, *n* = 6 mice in each group. Source data are provided as a Source data file.

---

properties of BCSCs including tumor initiation, progression, and self-renewal.

## KK-LC-1 targets FAT1 for proteasomal degradation in TNBC

We next investigated the molecular mechanism underneath the function of KK-LC-1 on the TNBC ALDH+ cells. RNA-seq analysis and KEGG pathway enrichment analysis were performed following *KK-LC-1* knockout[25]. The Hippo pathway, which regulates tissue-specific stem cells[26,27], was one of the top-ranked signaling pathways altered in *KK-LC-1*−/− MDA-MB-231 cells and *KK-LC-1*+/+ MDA-MB-231 cells (Fig. 3a). To reveal proteins that link KK-LC-1 with the Hippo pathway, proteins that could be immunoprecipitated with KK-LC-1 were quantified using mass spectrometry (Supplementary Data 2) and compared with known Hippo pathway components (Supplementary Data 2)[18]. FAT1 was identified as a high-confidence interacting protein (Fig. 3b). In drosophila, the atypical cadherin Fat (Ft) is a transmembrane protein that impacts Hippo signaling[28]. In humans, FAT1 regulates the Hippo pathway in ER+ breast cancer[29]. In mouse and human squamous cell carcinoma, FAT1 promotes tumor initiation, progression, invasiveness, stemness and metastasis[30]. Co-immunoprecipitation suggested direct molecular interaction between KK-LC-1 and FAT1 proteins in MDA-MB-231 cells and MDA-MB-468 cells (Fig. 3c, d). Western blotting showed that FAT1 protein levels were increased in *KK-LC-1*−/− MDA-MB-231 cells and *KK-LC-1*−/− MDA-MB-468 cells compared to parental MDA-MB-231 and MDA-MB-468 cells, and FAT1 protein levels were decreased in MDA-MB-231 cells overexpressing KK-LC-1 compared to negative control (NC) (Fig. 3e, f and Supplementary Fig. 4a). To confirm that KK-LC-1 regulates the expression of FAT1 at the protein level, qRT-PCR was used to measure the relative expression of *FAT1* mRNA in *KK-LC-1*−/− MDA-MB-231 and parental MDA-MB-231, with findings showing no significant differences (Supplementary Fig. 4b). Given that FAT1 is an enormous transmembrane protein, we wanted to examine if the expression of FAT1 can be regulated by modulating its protein degradation. Consequently, MDA-MB-231 cells were treated with different combinations of cycloheximide (protein synthesis inhibitor), MG132 (proteasome inhibitor), and bafilomycin (BAF) (lysosome inhibitor). Interestingly, we found that proteasome inhibitor MG132 could slow down the degradation of FAT1, indicating KK-LC-1 may regulate FAT1 via ubiquitination proteasomal pathway (Supplementary Fig. 4c, d). Furthermore, FAT1 protein levels were decreased after *KK-LC-1* overexpression in MDA-MB-231 cells or MDA-MB-468 cells and

this phenotype could be reversed by proteasome inhibitor MG132 rather than lysosome inhibitor BAF (Fig. 3g and Supplementary Fig. 4e). To ensure that KK-LC-1 could mediate the degradation of FAT1, we conducted CHX protein degradation experiments and we found that *KK-LC-1* knockout in MDA-MB-231 cells or MDA-MB-468 cells can significantly retard the degradation of FAT1 protein as compared to parental cells (Fig. 3h). Immunoprecipitation confirmed that *KK-LC-1* knockout led to a remarkable reduction in FAT1 ubiquitination compared to parental, and re-enforced expression of KK-LC-1 in *KK-LC-1*−/− MDA-MB-231 cells restored the ubiquitination of FAT1 (Fig. 3i and Supplementary Fig. 4f,g). Together, these results demonstrate that KK-LC-1 directly targets FAT1 for ubiquitination and proteasomal degradation thus regulating the expression of FAT1. Additionally, histological studies on KK-LC-1 and FAT1 expression in primary TNBC specimens showed that those two proteins were negatively correlated (Fig. 3j, k), suggesting the molecular interaction between KK-LC-1 and FAT1 proteins may have clinical relevance in TNBC.

## KK-LC-1-FAT1 protein−protein interaction maintains BCSC stemness through mediating the Hippo pathway

To ascertain that KK-LC-1 regulates stemness of ALDH+ cells via FAT1, we transduced *KK-LC-1*−/− MDA-MB-231 cells with *FAT1*-specific shRNA (knockdown efficacy, ~70%) (Supplementary Fig. 5a, b). The proportion of ALDH+ cells and ALDH+CD44highCD24low cells was significantly increased in *KK-LC-1*−/− MDA-MB-231 cells transduced with *FAT1*-specific shRNA compared to the negative control (NC) (Supplementary Fig. 5c, d).

As loss of *FAT1* can cause Hippo pathway suppression and YAP1 activation in breast cancer[29], proteins downstream of the Hippo pathway in TNBC were analyzed. The phosphorylation of MST1, LATS1, and YAP1 was increased in *KK-LC-1*−/− MDA-MB-231 cells and *KK-LC-1*−/− MDA-MB-468 cells compared to parental cells and knockdown of *FAT1* in *KK-LC-1*−/− MDA-MB-231 cells and *KK-LC-1*−/− MDA-MB-468 cells could deactivate Hippo signaling pathway (Fig. 4a, b). Furthermore, we found that *KK-LC-1* knockout can suppress the nuclear translocation of YAP1 and subsequently lead to the retention of YAP1 protein in the cytoplasm as compared to parental cells (Fig. 4c, d). These observations showed that increased FAT1 by *KK-LC-1* knockout activated Hippo signaling pathway. Key regulators of normal and cancer stem cells including the transcriptional factors OCT4, NANOG, and SOX2 were found significantly decreased in *KK-LC-1* knockout cells at both mRNA

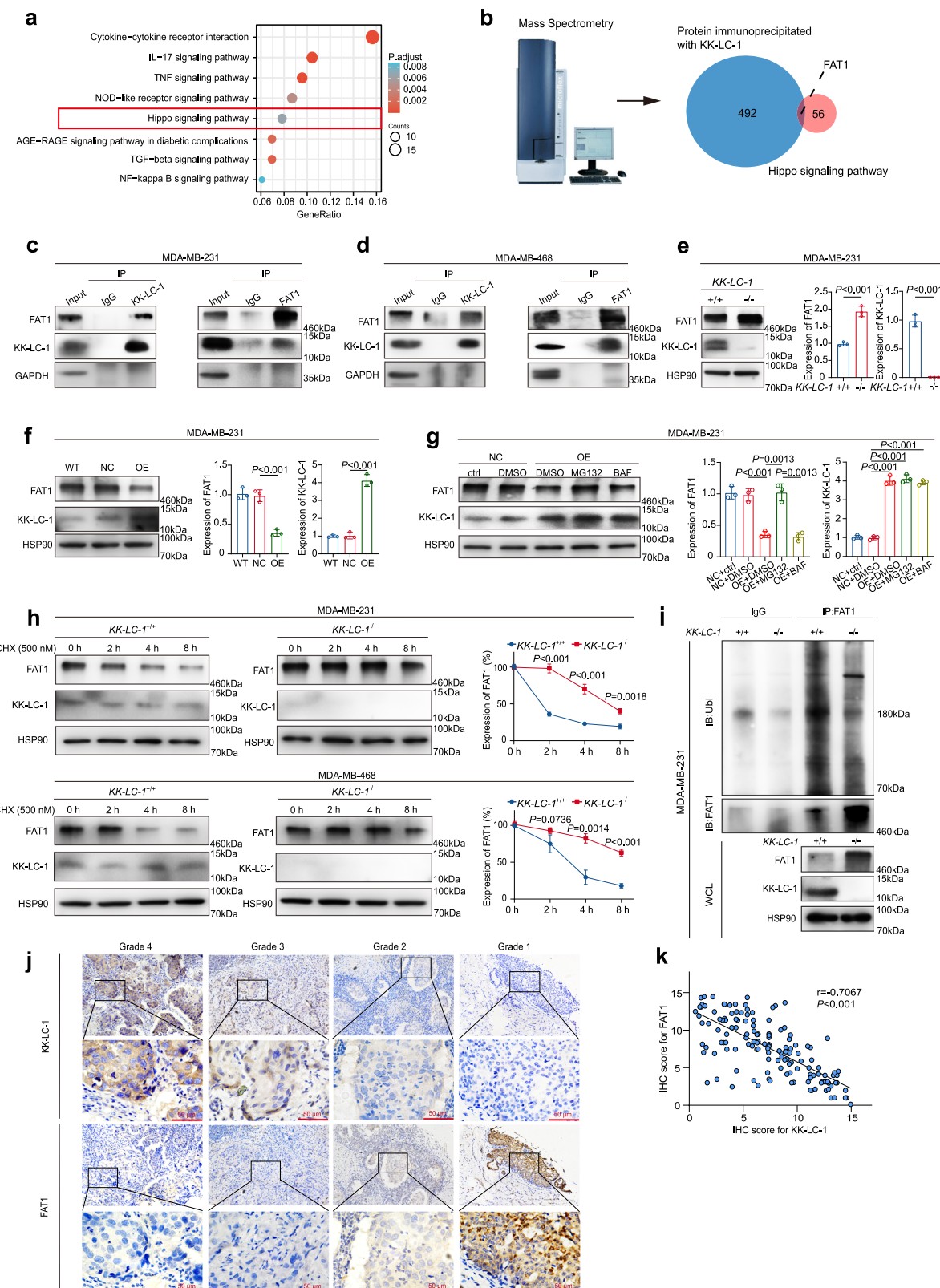

and protein levels (Fig. 4e, f and Supplementary Fig. 6a). Additionally, the key isoenzyme ALDH1A1 that is associated with ALDH⁺ stemness was also found significantly decreased at both mRNA and protein levels (Fig. 4e, f and Supplementary Fig. 6a). Smaller tumors derived from *KK-LC-1⁻ᐟ⁻* MDA-MB-231 cells in xenograft mice also had significantly reduced ALDH1A1 protein expression in comparison to the control tumors (Fig. 4g, h and Supplementary Fig. 6b). Similar to

*KK-LC-1⁻ᐟ⁻* MDA-MB-231 cells in vitro, FAT1 expression was increased in the *KK-LC-1⁻ᐟ⁻* tumors with decreased expression of stemness regulator SOX2 (Supplementary Fig. 6b). Together, our results implicate that KK-LC-1 can regulate ALDH⁺ stemness by mediating Hippo pathway through degrading FAT1. Moreover, the activation of Hippo pathway in turn inactivates YAP1 and diminishes the activities of the key isoenzyme ALDH1A1 and a few other stemness regulators (SOX2, OCT4,

**Fig. 3 | KK-LC-1 regulated FAT1 protein stability in TNBC. a** Differential gene enrichment analysis[25] showed that the Hippo pathway was one of the top signaling pathways altered in *KK-LC-1*[-/-] MDA-MB-231 cells and *KK-LC-1*[+/+] MDA-MB-231 cells, *P* values were determined using enrichKEGG function and adj *P* values were calculated using Benjamini and Hochberg's method. **b** FAT1 was identified as a high-confidence interacting protein with KK-LC-1. **c** Co-immunoprecipitation suggested direct molecular interaction between KK-LC-1 and FAT1 proteins in MDA-MB-231 cells, this experiment was performed 3 times independently with similar results. **d** Co-immunoprecipitation suggested direct molecular interaction between KK-LC-1 and FAT1 proteins in MDA-MB-468 cells, this experiment was performed 3 times independently with similar results. **e:** Western blotting showing FAT1 and KK-LC-1 protein levels in *KK-LC-1*[-/-] MDA-MB-231 cells and *KK-LC-1*[+/+] MDA-MB-231 cells, data are presented as mean ± SD from three biologically independent experiments, statistical significance was determined using Student's *t*-test and was two-sided. **f** Western blotting showing FAT1 and KK-LC-1 protein levels in MDA-MB-231 cells overexpressing KK-LC-1, negative controls, and parental MDA-MB-231 cells; data are presented as mean ± SD from three biologically independent experiments,

statistical significance was determined using Student's *t*-test and was two-sided. **g** Western blotting showing the effects of MG-132, bafilomycin, and overexpression of KK-LC-1 on FAT1 protein levels in MDA-MB-231 cells; data are presented as mean ± SD from three biologically independent experiments, statistical significance was determined using Student's *t*-test and was two-sided. **h** Western blotting showing degradation of FAT1 protein in *KK-LC-1*[-/-] MDA-MB-231 cells, *KK-LC-1*[-/-] MDA-MB-468 cells, *KK-LC-1*[+/+] MDA-MB-231 cells and *KK-LC-1*[+/+] MDA-MB-468 cells; data are presented as mean ± SD from three biologically independent experiments, statistical significance was determined using Student's *t*-test at each time point and was two-sided. **i** Immunoprecipitation showing ubiquitination of FAT1 protein in *KK-LC-1*[-/-] MDA-MB-231 cells and *KK-LC-1*[+/+] MDA-MB-231 cells, this experiment was performed 3 times independently with similar results. **j** Immunohistochemistry staining showing KK-LC-1 and FAT1 protein expression in TNBC tissue, *n* = 144 clinical samples, red scale bars indicate 50 μm. **k** Correlation between KK-LC-1 and FAT1 protein levels in TNBC tissue; *n* = 144 clinical samples, data are analyzed using Pearson correlation, *r* = −0.7067, *P* < 0.0001. Source data are provided as a Source data file.

and NANOG) within the TNBC BCSCs. Hence, our study may provide mechanistic insight for the development of potential therapeutics targeting BCSCs.

## Identification of small molecule compound that disrupts KK-LC-1/FAT1 protein-protein interaction

As a member of the cancer-testis antigens (CTAs) family[13], KK-LC-1 is barely expressed in normal tissue, except for the testis, and yet KK-LC-1 is overexpressed in tumor tissue. To this end, we want to identify small-molecule inhibitors that can disrupt the binding of KK-LC-1 with FAT1 via a computational approach. First, the structure of KK-LC-1 and FAT1 were built according to the structure information obtained from AlphaFold Protein Structure Database[31] and UniProtKB[32]. Subsequently, we conducted the protein-protein docking by ZDOCK followed by molecular dynamics simulations (MD) to optimize the complex structure of KK-LC-1/FAT1. Furthermore, we assessed the key amino acid residues at the complex interface to define the binding pocket of small-molecule compounds. Next, we employed virtual screening and the compounds were ranked according to the binding free energies at the complex interface by AutoDock Vina and our in-house developed tools FIPSDock (Supplementary Fig. 7a)[33]. Among the compounds, 145 small molecular compounds were predicted to bind in the pocket. The binding affinity to KK-LC-1 protein and $K_D$ of each drug were detected using biolayer interference (BLI) (Fig. 5a and Supplementary Fig. 7b-8). Z839878730 (Z8) was selected as the candidate molecule with the best affinity to KK-LC-1, with a $K_D$ of 4.3 × $10^{-6}$ mol/L (Fig. 5b, c). A KK-LC-1-Laminin G heterodimer receptor system was constructed with an interface area of 734.7 Å, solvation-free energy of −6.8 kcal/mol and binding free energy of −21.32 kcal/mol (mm/GBSA model). A comparison between the initial conformation of the KK-LC-1-Laminin G heterodimer receptor system and the last frame of the molecular dynamics (MD) simulations showed reasonable agreement (Fig. 5d). Molecular docking detected several hot-spot residues that contributed to the interaction between Z8 and KK-LC-1. These hot-spot residues were part of beta-sheets and their flanking strands. High-interest hot-spot residues (center_x = 36.294; center_y = 49.379; center_z = 37.191) were selected for analysis. Strong hydrogen bonding was identified between LEU-94 and GLU-95 on KK-LC-1, GLN-131, and HIS-170 on the Laminin G domain and Z8 (Fig. 5e). Weaker (binding free energy, −9.2 kcal/mol) hydrogen bonding and hydrophobic interactions were identified between LEU-94 and GLU-95 on KK-LC-1 and Z8, and LEU-91, LEU-98, GLU-93, and THR-97 on KK-LC-1, TYR-168, LEU-16, GLN-130, TYR-121, ILE-128, GLN-131 and HIS-170 on the Laminin G domain and Z8, respectively (Fig. 5e). To investigate the interaction between the KK-LC-1-Laminin G heterodimer receptor system and Z8, a KK-LC-1-Laminin G-Z8 complex was built. After 100 ns of MD simulations, Z8 significantly disrupted the interaction between

KK-LC-1 and the Laminin G domain (Fig. 5f). The small molecule Z8 occupied the groove consisting of beta-sheets and several loop structures, disrupting the interaction between the C-terminal domain of KK-LC-1 and the Laminin G domain.

In particular, to assess the putative capability of small-molecule compound Z8 disrupting the KK-LC-1 and FAT1 binding, we further performed the steered molecular dynamics (SMD) simulations on the equilibrated model after the 100 ns MD simulations. A constant pulling velocity was applied to KK-LC-1 to pull it away from the binding surface of the Laminin G domain. Snapshots of the structure were captured every 200 ps and steering forces were recorded every 50 ps. The models were pulled for 1 ns and during the 1 ns SMD simulations (Supplementary Fig. 9), we found that KK-LC-1 can easily depart from FAT1 in Z8-KK-LC-1/Laminin G complex as compared to KK-LC1−FAT1 complex. The distances between the center of mass of the KK-LC-1 and that of FAT1 before and after pulling were 1.953 and 2.953 nm in KK-LC1−FAT1 complex, whereas the distances were 1.259 and 2.259 nm in Z8-KK-LC1−Laminin G complex. These results implicated that Z8 may be able to disrupt the binding between KK-LC-1 to FAT1.

To detect the biological activity of Z8, half maximal inhibitory concentrations (IC$_{50}$) were calculated. IC$_{50}$ values of Z8 on MDA-MB-231 cells and MDA-MB-468 cells were 9.7 μM and 13.67 μM, respectively (Supplementary Fig. 10a). Z8 showed selectivity and had little effect on MCF7, SKBR3, and MCF10A cells, which barely expressed KK-LC-1 (Supplementary Fig. 10a). Besides, Z8 displayed little effect on *KK-LC-1*[-/-] MDA-MB-231 cells and *KK-LC-1*[-/-] MDA-MB-468 cells (Supplementary Fig. 10b). In docetaxel-resistant MDA-MB-231 cells, Z8 combined with docetaxel exerted synergistic effect, suggesting Z8 could reduce the drug resistance of docetaxel-resistant MDA-MB-231 cells to docetaxel (Supplementary Fig. 10c). To confirm if Z8 can disrupt the binding of KK-LC-1 and FAT1, we first performed a co-immunoprecipitation study on MDA-MB-231 cells and MDA-MB-468 cells following in vitro Z8 treatment. At the concentration of 5 μM, Z8 dramatically decreased the co-immunoprecipitation of FAT1 and KK-LC-1 compared to DMSO treatment (Fig. 5g and Supplementary Fig. 10e). Interestingly, we found that FAT1 protein degradation was retarded by Z8 compound at the same concentration (Fig. 5h and Supplementary Fig. 10d), in addition to the suppression of FAT1 ubiquitination in MDA-MB-231 cells (Fig. 5i). Moreover, MST1, LATS1, and YAP1 phosphorylation was increased in a dose-dependent manner in MDA-MB-231 cells and MDA-MB-468 cells treated with Z8, indicating activation of the Hippo signaling pathway downstream of FAT1 (Fig. 5j and Supplementary Fig. 10f). Furthermore, immuno-fluorescence staining of YAP1 in MDA-MB-231 cells and MDA-MB-468 cells also showed that the treatment of 5 μM Z8 significantly impaired the nuclear shift of YAP1 compared to DMSO treatment (Supplementary Fig. 10g,h). Hence, the identification of Z8 targeting

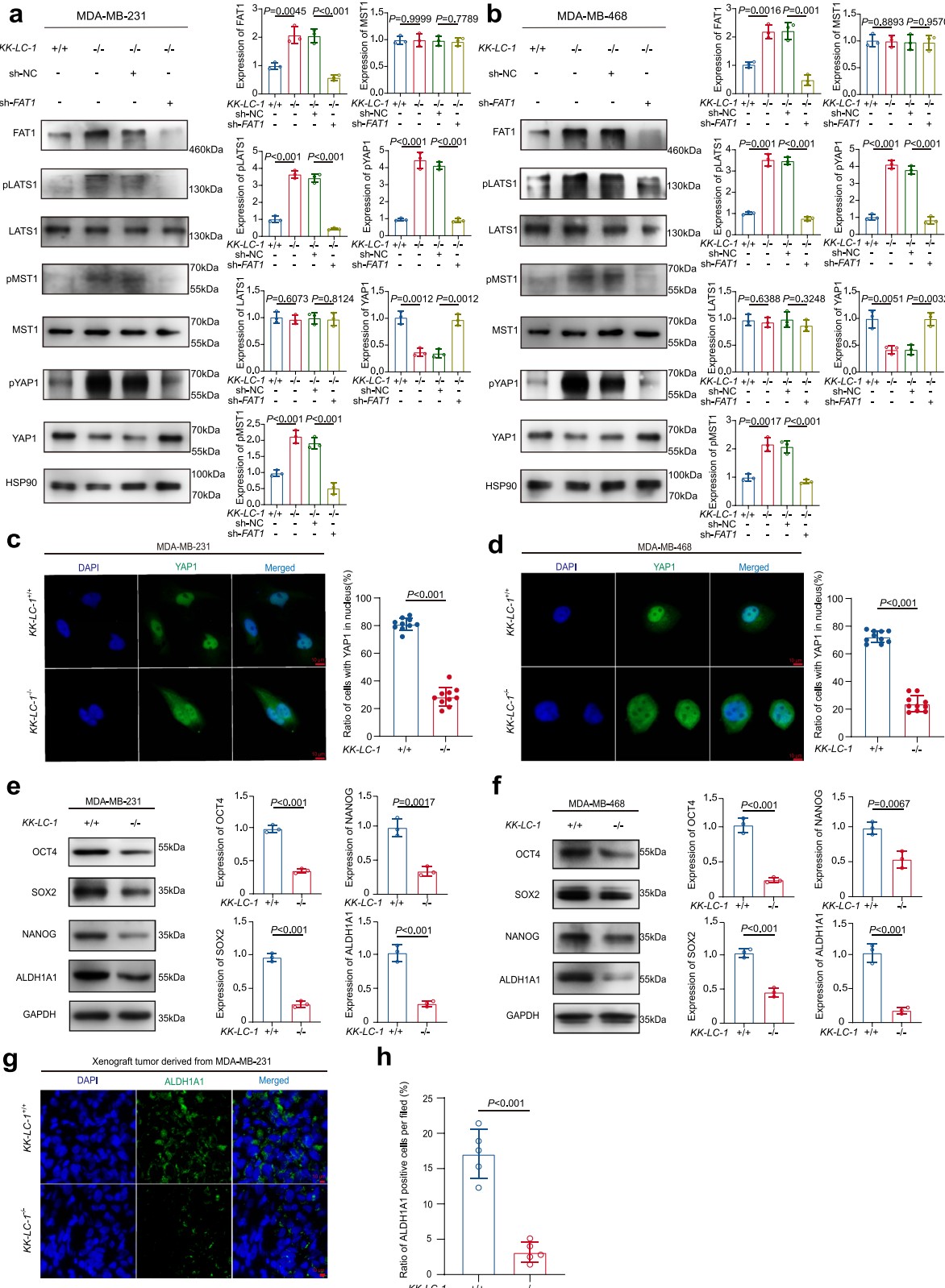

the KK-LC-1–FAT1–Hippo axis may open avenues for developing a pharmacological treatment for drug-resistant TNBC.

## Z8 impairs ALDH⁺ cells and inhibits tumor growth in vivo

To explore whether interrupting KK-LC-1–FAT1 interaction by the small molecule Z8 possesses potent anti-BCSC activity, both MDA-MB-231 cells and MDA-MB-468 cells were cultured in the presence of Z8. In vitro Z8 treatment dramatically reduced the proportion of ALDH⁺ cells

in a dose-dependent manner (Fig. 6a, b). The sphere-forming ability of ALDH⁺ cells was also significantly impaired by the treatment of Z8, manifested by both decreased number and smaller diameter of tumor spheres than the DMSO control spheres (Fig. 6c, d). In contrast, the Z8 compound exerted little effect in ALDH⁻ cells with minimal sphere formation ability (Fig. 6c, d). The self-renewal ability of ALDH⁺ MDA-MB-231 cells was also impaired with the treatment of Z8, detected by sphere formation assay under limiting dilution conditions

**Fig. 4 | KK-LC-1 knockout maintained the Hippo pathway in an active state via FAT1. a** Western blotting showing FAT1, LATS1, pLATS1, MST1, pMST1, YAP1, and pYAP1 protein levels in *KK-LC-1*[+/+] MDA-MB-231 cells, *KK-LC-1*[−/−] MDA-MB-231 cells, *KK-LC-1*[−/−] MDA-MB-231 cells transduced with negative control lentivirus and *KK-LC-1*[−/−] MDA-MB-231 cells transduced with sh-*FAT1* lentivirus; data are presented as mean ± SD from three biologically independent experiments, statistical significance was determined using Student's *t*-test and was two-sided. **b** Western blotting showing FAT1, LATS1, pLATS1, MST1, pMST1, YAP1, and pYAP1 protein levels in *KK-LC-1*[+/+] MDA-MB-468 cells, *KK-LC-1*[−/−] MDA-MB-468 cells, *KK-LC-1*[−/−] MDA-MB-468 cells transduced with negative control lentivirus and *KK-LC-1*[−/−] MDA-MB-468 cells transduced with sh-*FAT1* lentivirus; data are presented as mean ± SD from three biologically independent experiments, statistical significance was determined using Student's *t*-test and was two-sided. **c** Immunofluorescence showing the distribution of YAP1 protein in *KK-LC-1*[−/−] MDA-MB-231 cells and *KK-LC-1*[+/+] MDA-MB-231 cells, red scale bars indicate 10 μm; data are presented as mean ± SD from 10 randomized views and this experiment was performed independently for 3 times, statistical significance was determined using Student's *t*-test and was two-sided. **d** Immunofluorescence showing the distribution of YAP1 protein in *KK-LC-1*[−/−] MDA-MB-468 cells and *KK-LC-1*[+/+] MDA-MB-468 cells, red scale bars indicate 10 μm; data are presented as mean ± SD from 10 randomized views and this experiment was performed independently for 3 times, statistical significance was determined using Student's *t*-test and was two-sided. **e** Western blotting showing SOX2, OCT4, NANOG, and ALDH1A1 protein levels in *KK-LC-1*[−/−] MDA-MB-231 cells and *KK-LC-1*[+/+] MDA-MB-231 cells; data are presented as mean ± SD from three biologically independent experiments, statistical significance was determined using Student's *t*-test and was two-sided. **f** Western blotting showing SOX2, OCT4, NANOG, and ALDH1A1 protein levels in *KK-LC-1*[−/−] MDA-MB-468 cells and *KK-LC-1*[+/+] MDA-MB-468 cells; data are presented as mean ± SD from three biologically independent experiments, statistical significance was determined using Student's *t*-test and was two-sided. **g** and **h** Immunofluorescence showing ALDH1A1 protein expression in tumors derived from *KK-LC-1*[−/−] MDA-MB-231 cells and *KK-LC-1*[+/+] MDA-MB-231 cells, red scale bars indicate 10 μm; data are presented as mean ± SD from five randomized views and this experiment was performed independently for 3 times, statistical significance was determined using Student's *t*-test and was two-sided. Source data are provided as a Source data file.

(Supplementary Table 3). Decreased expression of BCSC stemness regulators ALDH1A1 and SOX2 were observed in the tumor spheres treated with Z8 (Fig. 6e, f). To test if Z8 can inhibit tumor growth in vivo, nude mice inoculated with MDA-MB-231 cells were treated with various doses of Z8. Z8 significantly inhibited tumor growth in a dose-dependent manner (Fig. 6g–i).

To mimic the characteristics of patients with TNBC, PDX mouse models were established using clinical samples that highly expressed KK-LC-1. Mice were randomly assigned into different groups and treated with docetaxel, Z8, or a combination of both. Z8 alone had a much better inhibitory effect on tumor growth than docetaxel alone or DMSO vehicle treatment. The combination of docetaxel and Z8 had a best inhibitory effect than each single drug treatment, indicating that Z8 synergizes with the chemotherapeutic drug docetaxel to promote the efficacy on TNBC with KK-LC-1 overexpression (Fig. 7a–c). Notably, we revealed that Z8 exerted the anti-tumor effects in the PDX animal models via mediating the KK-LC-1–FAT1–Hippo axis. We found that FAT1 was significantly increased in Z8-treated xenografts in addition to the retention of YAP1 in the cytoplasm of tumor cells and suppression of ALDH1A1 (Fig. 7d, e). Therefore, these results collectively demonstrated that the small molecule Z8 exerts great anti-TNBC activity by activating the Hippo pathway within the ALDH[+] cells.

### Small molecule compound Z8 displays a safety profile
To determine the suitability of small molecule Z8 in clinical practice, we assessed the acute toxic effects of Z8 in mice. Briefly, thirty C57 mice were exposed to intraperitoneal administration of Z8 at five different doses over the course of one day whereas another six mice were exposed to vehicle only (normal saline containing 5% DMSO and 1% carboxymethylcellulose, *n* = 6). The Z8-treated animals (dose under 500 mg/kg) did not show any signs of lethargy, weight loss, or other physical indications of sickness. The mice were sacrificed after a 14-day washout period to assess delayed toxicity. Heart, liver, lung, and kidney were examined and no toxic effects or tissue damage (either macroscopic or microscopic) were observed (Supplementary Fig. 11a). Finally, we determined the LD$_{50}$ of Z8 is about 1300 mg/kg (Supplementary Fig. 11b), implicating its safety profile in animals.

## Discussion
To date, conventional chemotherapy or radiation therapy is unable to eradicate BCSCs[34]. Accumulating evidence suggests that BCSCs are drivers of relapse and recurrence of TNBC[35]. Moreover, BCSCs play a central role in drug resistance and immune surveillance, which facilitates tumor growth[35,36]. Hence, therapeutics that can ablate BCSCs are urgently needed in clinical practice.

ALDH[+] cells represent a potential target for CSC-directed therapy in TNBC[37]. ALDH[+] cells are considered a key subpopulation responsible for metastatic seeding and outgrowth, and ALDH[+] cells have epithelial-like properties[12], a distinct EMT gene expression profile[10], and enhanced tumorigenicity in vivo. Among cancer stem cell subgroups, the ALDH[+] subgroup is considered to be more prone to create colonies, have greater tumorigenic potential, and are more drug resistant[9,12].

Several proteins or transcription factors that can positively regulate ALDH[+] TNBC cells have been reported in previous studies, including PDE5[38], SOX2[39], BMI1[40], eIF4A[41], and Notch[42–45]. However, small molecule inhibitors or monoclonal antibodies targeting these upstream regulators have not yet been developed. Although some inhibitors that can inhibit Notch1, such as Psoralidin[43], have been identified, their mechanism of action has not been elaborated in relevant studies. In addition, the Wnt/β-catenin and Hippo/YAP pathways have also been found to regulate the number of ALDH[+] cells in TNBC[46]. In a human xenograft model, both CD44[high]CD24[low] and ALDH[+] BCSC subpopulations were reduced following the dual administration of the Wnt inhibitor ICG-001 and the YAP inhibitor simvastatin, suggesting a therapeutic strategy for TNBC through the dual administration of Wnt and YAP[46]. However, given the important role of Wnt/β-catenin and Hippo/YAP pathways for normal cells[47,48], direct inhibition of Wnt/β-catenin and Hippo/YAP pathways may trigger significant side effects.

This study explored the underlying mechanisms controlling the proportion of TNBC ALDH[+] cells and ALDH-associated stemness. We demonstrate that KK-LC-1 is important for the maintenance of TNBC BCSCs, especially ALDH[+] cells and that KK-LC-1 controls the stemness of TNBC ALDH[+] cells via binding with FAT1, promoting FAT1 ubiquitination and degradation, and compromising Hippo signaling, which resulted in nuclear translocation of YAP1 and ALDH1A1 transcription. Noteworthy, we reported the identification of a small molecule compound, named as Z8, which can be home at the protein–protein interface of KK-LC-1/FAT complex. We revealed that the Z8 compound enables the reactivation of the Hippo pathway and subsequent suppression of TNBC ALDH[+] cell stemness and viability via disrupting the binding of KK-LC-1 to FAT1. To date, targeting protein–protein interaction is still a rather challenging task in drug discovery campaigns. The key point of drug design to target protein–protein interactions is to find key amino acid residues at the site of action[49]. Furthermore, capturing the dynamic conformation of protein-protein interactions for virtual screening of compound library is one prerequisite for the design of protein–protein inhibitors (PPI). In the present work, we employed molecular dynamics simulations to optimize the complex structure of KK-LC-1 and FAT1. Moreover, key amino acid residues at the binding interface of KK-LC-1 and FAT1 were evaluated to define the pocket of compound virtual screening. Several small molecule drugs

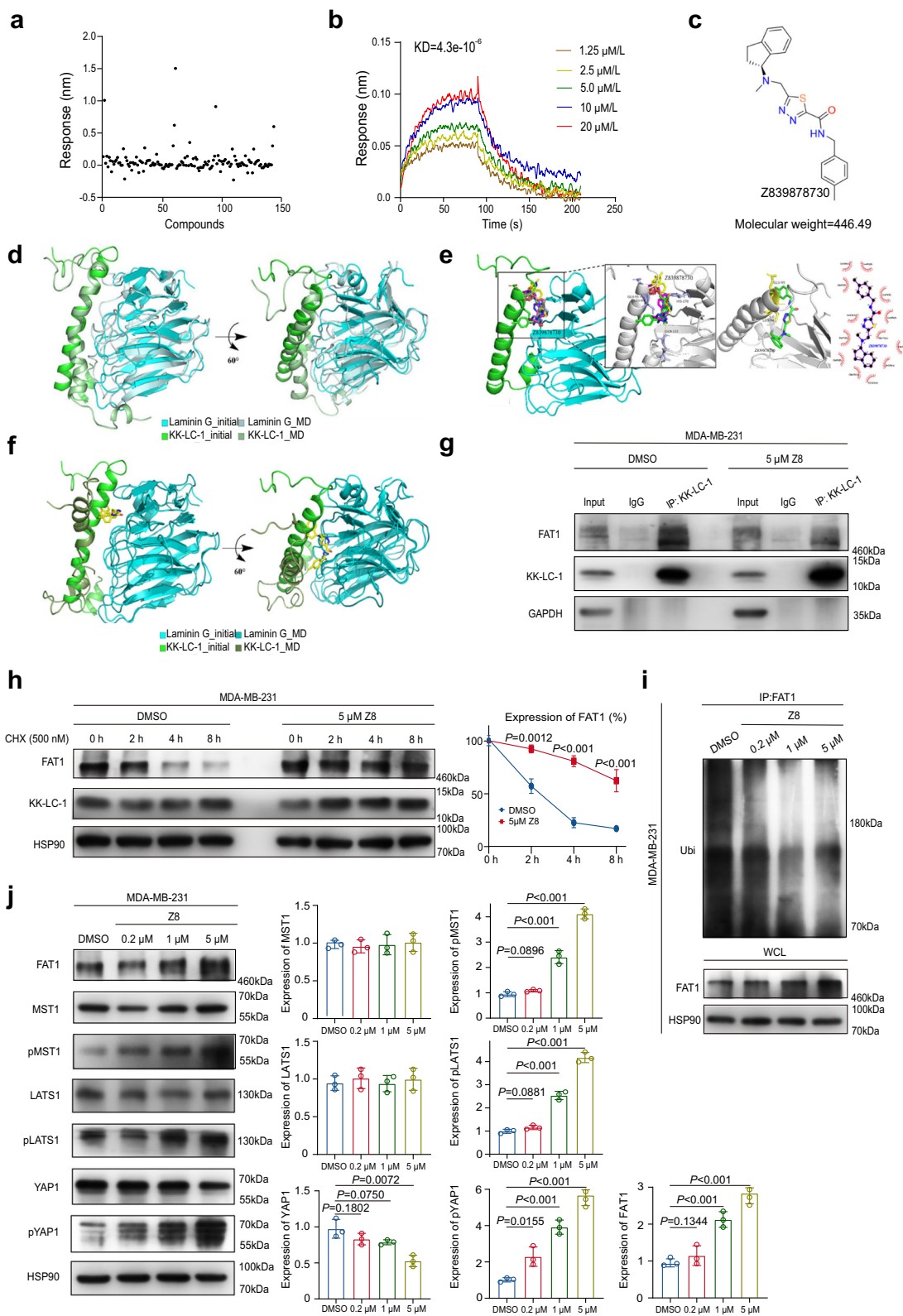

identified using computational methods have been applied to breast cancer treatment, including alpelisib, a phosphatidylinositol-3 kinase alpha inhibitor[50], talazoparib tosylate, a poly (ADP-ribose) polymerase inhibitor[51], neratinib maleate, a receptor tyrosine kinase, human epidermal growth factor receptor 2, and human EGFR inhibitor, and ribociclib, a cyclin-dependent kinase (CDK) inhibitor. Further preclinical and clinical work is ongoing to optimize Z8 as a strategy in TNBC therapy.

A previous study demonstrated that hypomethylation-induced KK-LC-1 facilitated HCC progression, and KK-LC-1 physically interacted with presenilin-1 to promote the Notch1/Hes1 pathway in HCC[16]. We identified the KK-LC-1–FAT1–Hippo–YAP–ALDH1A1 pathway in BCSCs as an oncogenic signaling pathway in TNBC. KK-LC-1 is a cancer germline antigen that is rarely expressed in normal tissue but abundantly expressed in malignant foci, especially epithelial cancers, including lung cancer, gastric cancer, and breast cancer[13,52,53]. The overexpression of

**Fig. 5 | Z839878730 (Z8) disrupted KK-LC-1-FAT1 protein-protein interactions and prevented FAT1 degradation. a** 145 small molecular compounds were predicted to having binding affinity with KK-LC-1 and their binding signals with KK-LC-1 protein were tested. **b** Multi-concentration association-dissociation curve of Z8 and KK-LC-1 protein interaction, KD = 4.3 × 10⁻⁶ mol/L. **c** Structure of Z8. **d** Comparison between the initial conformation of the KK-LC-1-laminin G heterodimer receptor system and the last frame of the MD simulations[31,32]. **e** Hot-spot residues that contributed to the interaction between Z8 and KK-LC-1. Yellow lines radiating towards the ligands are hydrogen binding. Arcs with spokes radiating toward the ligands are hydrophobic interactions[31,32]. **f** Comparison between the initial conformation of the KK-LC-1-laminin G-Z8 complex and the last frame of the MD simulations[31,32]. **g** Co-immunoprecipitation of FAT1 and KK-LC-1 was reduced by treatment of MDA-MB-231 cells with 5 μM Z8 compared to DMSO, this experiment was performed 3 times independently with similar results. **h** Western blotting showing degradation of FAT1 was significantly decreased by treatment of MDA-MB-231 cells with 5 μM Z8 compared to DMSO; data are presented as mean ± SD from three biologically independent experiments, statistical significance was determined using Student's t-test and was two-sided. **i:** Immunoprecipitation showing ubiquitination of FAT1 was decreased by treatment of MDA-MB-231 cells with various concentrations of Z8 compared to DMSO, this experiment was performed 3 times independently with similar results. **j** Western blotting showing FAT1, MST1, pMST1, LATS1, pLATS1, YAP1, and pYAP1 protein levels in MDA-MB-231 cells treated with different concentrations of Z8 or DMSO; data are presented as mean ± SD from three biologically independent experiments, statistical significance was determined using Student's t-test and was two-sided. Source data are provided as a Source data file.

KK-LC-1 in patients with TNBC[54] yet without expression in normal tissues enables KK-LC-1 as a viable target for the development of therapeutic agents against TNBC with minimal side effects. Consistent with this, T cell receptor (TCR) gene therapy targeting KK-LC-1 has achieved considerable efficacy in lung cancer and HCC.

FAT1 is frequently mutated in human cancer and acts as an oncogene or tumor suppressor depending on the type of cancer. In breast cancer, decreased FAT1 expression has been associated with high histological grade, poor lymph node status, progression, aggressive behavior, and poor prognosis[55,56]. In addition, overexpression of FAT1 reduces stem cell markers and inhibits spheroid formation in non-small cell lung cancer (NSCLC) cells, and FAT1 may reduce tumor initiation in NSCLCs by promoting nucleoplasmic translocation of YAP1[57]. The Hippo pathway also plays a crucial role in cancer[58]. The Hippo pathway includes the kinases MST1/2 and LATS1/2 that phosphorylate the cotranscription factors YAP1 and TAZ[59]. In breast cancer, dysregulation of the Hippo pathway facilitates tumorigenesis and metastasis by promoting EMT, stem cell generation, and therapeutic resistance[59,60]. In TNBC, YAP1 and TAZ expression are important drivers of malignancy through interaction with Aurora kinase A, a regulator of mitotic cell proliferation[61], and TAZ may be a predictive biomarker for drug response[62]. FAT1 loss-of-function mutations have been associated with drug resistance in patients with ER⁺ breast cancer treated with CDK 4/6 inhibitors via a mechanism involving the Hippo pathway and changes in gene expression caused by YAP1[29].

Besides, there is cross-talk between the Hippo pathway and the Wnt/β-catenin, TGF-β, PI3K/AKT, MAPK, JAK/STAT, EGFR, and JNK pathways[59]. FAT1 is able to regulate tumor cell stemness through Hippo-independent pathways. Morris et al. reported that endogenous FAT1 binds to β-catenin in human cells and affects Wnt signaling by enhancing β-catenin activity[63]. Loss of FAT1 can lead to dysregulated Wnt signaling and breast tumorigenesis. In ductal carcinoma in situ (DCIS) and invasive ductal carcinoma (IDC), FAT1 and β-catenin expression are positively correlated with each other, and FAT1(−), β-catenin(−) or FAT1(−)/β-catenin(−) suggest worse disease-free survival[55]. FAT1 knockdown may enhance stemness and ABCC3-associated cisplatin resistance in esophageal cancer cells via the Wnt/β-linked protein signaling pathway[64].

In conclusion, our study reveals that KK-LC-1 regulates ALDH1 expression that indirectly mediates TNBC progression. Targeting KK-LC-1 with a small molecule compound (Z8) eliminated ALDH⁺ cells and achieved tumor regression, implicating KK-LC-1 as a novel therapeutic target in TNBC. Our findings provide preclinical evidence of defining Z8 compounds as a promising therapeutic agent which is capable of disrupting KK-LC-1/FAT1 to deplete BCSCs for the treatment of TNBC.

## Methods

This research complies with all relevant ethical regulations including human clinical samples and animal studies. The study protocol was approved by the Ethics Committee of the China Medical University.

### Reagents or animal models

**mRNA seq data generation.** The total mRNA of docetaxel-resistant MDA-MB-231 cells and its parental cells were collected using RNAiso Plus (9108, Takara) (3 replicates for each group), cryopreserved in drikold and subjected to Biomarker Technologies and then sequenced using Illumina platform. The total mRNA of *KK-LC-1⁺/⁺* MDA-MB-231 cells and *KK-LC-1⁻/⁻* MDA-MB-231 cells were also collected using RNAiso Plus (9108, Takara) (3 replicates for each group), cryopreserved in drikold and subjected to Bgi Genomics Co., Ltd and then sequenced using Illumina platform.

**mRNA seq data analysis.** Raw RNA-sequencing count data to analyze the TNBC-specific up-regulated genes were downloaded from TCGA Breast Cancer (TCGA-BRCA) dataset (https://portal.gdc.cancer.gov/)[19]. Molecular and clinical data for the TNBC samples were obtained from a previous publication[20].

Limma and ggplot2 R package were used to perform the differential expressed genes analysis. Gene expression in 194 TNBC tissue samples and 113 adjacent normal breast tissue samples in the TCGA database were analyzed[20]; P values were determined using moderated t-statistic and adj P values were calculated using Benjamini and Hochberg's method. Cut-off criteria were Log2 Fold change ≥ 6 and adj P-value < 0.05. *KK-LC-1* expression was investigated in 1102 breast cancer tissue samples and 113 para-tumor tissue samples. The Kaplan Meier plotter (http://kmplot.com) was used for the analysis of *KK-LC-1* mRNA expression and clinical outcomes[21]. The mRNA expression data of *KK-LC-1* in various breast cancer cell lines were obtained from Cancer Cell Line Encyclopedia (CCLE) (https://sites.broadinstitute.org/ccle/tools)[22]. Kyoto Encyclopedia of Genes and Genomes (KEGG) pathway enrichment[25] was performed using the clusterProfiler R package, P values were determined using the enrichKEGG function, and adj P values were calculated using Benjamini and Hochberg's method.

**Breast cancer patients and tissue samples.** A total of 144 female patients (Supplementary Table 4) with invasive ductal breast cancer with the triple-negative phenotype were recruited from the breast centres affiliated with the China Medical University, Shenyang, China. Tissues were collected from primary breast tumors during surgery. All patients received standardized post-operative adjuvant therapy according to National Comprehensive Cancer Network (NCCN) guidelines (v 1. 2008) Inclusion criteria were: (1) complete clinical data; (2) minimum follow-up period of ≥5 years; and (3) underwent axillary lymph node or sentinel lymph node dissection. Exclusion criteria were (1) incomplete clinicopathological data; (2) received any neoadjuvant therapy; (3) refused to receive standard adjuvant therapy; and (4) unknown survival status. The protocol was approved by the Ethics Committee of China Medical University (#2020PS106K[X1]). All patients provided written informed consent.

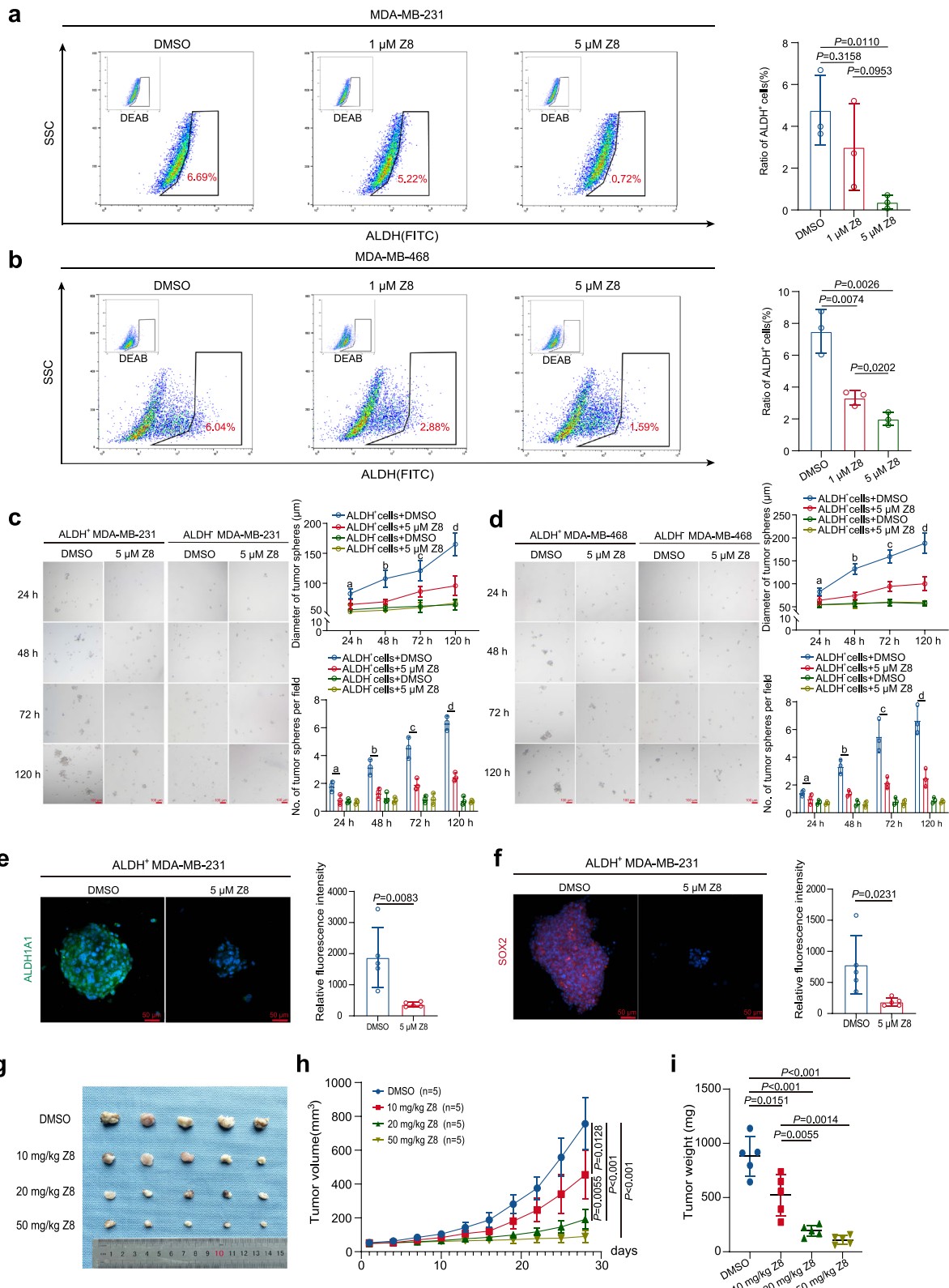

## Xenograft studies

Four to five-week-old female athymic BALB/c nu-mice which originated from Charles River Japan(CRJ) were maintained in the animal husbandry facility of a specific pathogen-free (SPF) laboratory in Shengjing Hospital of China Medical University. The day and night cycle in the SPF laboratory was 12 h/12 h, with the temperature at 20–24 °C and humidity at 40–70%. All experiments were performed in accordance with the Regulations for the Administration of Affairs Concerning

Experimental Animals and were approved by the Experimental Animal Ethics Committee of China Medical University (2020PS318K). These mice were used for tumor propagation assay and all the in vivo experiments that require tumor cell inoculation.

Intradermal injections of $1 \times 10^6$ *KK-LC-1*$^{-/-}$ MDA-MB-231 cells or *KK-LC-1*$^{+/+}$ MDA-MB-231 cells were performed to induce tumors. 2 weeks after tumor cell inoculation, tumor volume was measured every 3 days and calculated as $V = 1/2$ (width$^2$ × length).

**Fig. 6 | Z8 impaired ALDH+ cells and inhibited tumor growth in vivo.**
**a** Proportion of ALDH+ cells in MDA-MB-231 cells treated with different concentrations of Z8; data are presented as mean ± SD from three biologically independent experiments, statistical significance was determined using Student's *t*-test and was two-sided. **b** Proportion of ALDH+ cells in MDA-MB-468 cells treated with different concentrations of Z8; data are presented as mean ± SD from three biologically independent experiments, statistical significance was determined using Student's *t*-test and was two-sided. **c** Sphere formation by ALDH+ MDA-MB-231 cells and ALDH- MDA-MB-231 cells treated with 5 μM Z8 or DMSO, red scale bars indicate 100 μm, sphere diameter and a number of spheres were analyzed; data are presented as mean ± SD from three biologically independent experiments. Regarding to sphere diameter, 'a' indicates $P = 0.0486$ when ALDH+ MDA-MB-231 cells + Z8 compared to ALDH+ MDA-MB-231 cells + DMSO at 24 h time point, 'b' indicates $P = 0.0099$ when ALDH+ MDA-MB-231 cells + Z8 compared to ALDH+ MDA-MB-231 cells + DMSO at 48 h time point, 'c' indicates $P = 0.0324$ when ALDH+ MDA-MB-231 cells + Z8 compared to ALDH+ MDA-MB-231 cells + DMSO at 72 h time point, 'd' indicates $P = 0.0087$ when ALDH+ MDA-MB-231 cells + Z8 compared to ALDH+ MDA-MB-231 cells + DMSO at 120 h time point; Regarding to sphere number, 'a' indicates $P = 0.0292$ when ALDH+ MDA-MB-231 cells + Z8 compared to ALDH+ MDA-MB-231 cells + DMSO at 24 h time point, 'b' indicates $P = 0.0035$ when ALDH+ MDA-MB-231 cells + Z8 compared to ALDH+ MDA-MB-231 cells + DMSO at 48 h time point, 'c' indicates $P = 0.0060$ when ALDH+ MDA-MB-231 cells + Z8 compared to ALDH+ MDA-MB-231 cells + DMSO at 72 h time point, 'd' indicates $P < 0.001$ when ALDH+ MDA-MB-231 cells + Z8 compared to ALDH+ MDA-MB-231 cells + DMSO at 120 h time point; statistical significance was determined using Student's *t*-test and was two sided. **d** Sphere formation by ALDH+ MDA-MB-468 cells and ALDH- MDA-MB-468 cells treated with 5 μM Z8 or DMSO, red scale bars indicate 100 μm, sphere diameter and number of spheres were analyzed; data are presented as mean ± SD from three biologically independent experiments, Regarding to sphere diameter, 'a' indicates $P = 0.0566$ when ALDH+ MDA-MB-468 cells + Z8 compared to ALDH+ MDA-MB-468 cells + DMSO at 24 h time point, 'b' indicates $P = 0.0019$ when ALDH+ MDA-MB-468 cells + Z8 compared to ALDH+ MDA-MB-468 cells + DMSO at 48 h time point, 'c' indicates $P = 0.0026$ when ALDH+ MDA-MB-468 cells + Z8 compared to ALDH+ MDA-MB-468 cells + DMSO at 72 h time point, 'd' indicates $P = 0.0045$ when ALDH+ MDA-MB-468 cells + Z8 compared to ALDH+ MDA-MB-468 cells + DMSO at 120 h time point; Regarding to sphere number, 'a' indicates $P = 0.1458$ when ALDH+ MDA-MB-468 cells + Z8 compared to ALDH+ MDA-MB-468 cells + DMSO at 24 h time point, 'b' indicates $P = 0.0035$ when ALDH+ MDA-MB-468 cells + Z8 compared to ALDH+ MDA-MB-468 cells + DMSO at 48 h time point, 'c' indicates $P = 0.0098$ when ALDH+ MDA-MB-468 cells + Z8 compared to ALDH+ MDA-MB-468 cells + DMSO at 72 h time point, 'd' indicates $P = 0.0041$ when ALDH+ MDA-MB-468 cells + Z8 compared to ALDH+ MDA-MB-468 cells + DMSO at 120 h time point; statistical significance was determined using Student's *t*-test and was two sided. **e** Immunofluorescence showing ALDH1A1 protein expression in spheres formed by ALDH+ MDA-MB-231 cells treated with 5 μM Z8 or DMSO, red scale bars indicate 50 μm; data are presented as mean ± SD from five randomized views, this experiment was performed 3 times independently with similar results, statistical significance was determined using Student's *t*-test and was two-sided.
**f** Immunofluorescence showing SOX2 protein expression in spheres formed by ALDH+ MDA-MB-231 cells treated with 5 μM Z8 or DMSO, red scale bars indicate 50 μm; data are presented as mean ± SD from five randomized views, this experiment was performed 3 times independently with similar results, statistical significance was determined using Student's *t*-test and was two-sided. **g** Outcomes of tumors derived from MDA-MB-231 cells in mice treated with different doses of Z8, $n = 5$ mice for each group. **h** Volume of tumors derived from MDA-MB-231 cells in mice treated with different doses of Z8; data are presented as mean ± SD, $n = 5$ mice for each group, statistical significance was determined using Student's *t*-test and was two-sided. **i** Weight of tumors derived from MDA-MB-231 cells in mice treated with different doses of Z8, data are mean ± SD; data are presented as mean ± SD, $n = 5$ mice for each group, statistical significance was determined using Student's *t*-test and was two-sided. Source data are provided as a Source data file.

As for the drug sensitivity tests, small molecule compound Z8 or doctaxel was administrated when tumor volumes reached nearly 50 mm³. Mice were randomly assigned to one of four groups ($n = 5$ each). Mice carried xenograft tumor derived from cell line or xenograft tumor derived from TNBC patients were treated by intraperitoneal injection for 28 days with vehicle (1% DMSO dissolved in normal saline/2d), Z8 (10, 20, 50 mg/kg/2 day), docetaxel (10 mg/kg/1/2 week), or a combination of both drugs. Tumor volume measurements were performed by independent researchers who were blinded as to treatment group assignment. Mice were humanely euthanized (cervical dislocation under anesthesia using isoflurane) at the end of the experiment with maximal tumor volume <2000 mm³ and maximal weight loss <20% of their body weight. Mice were humanely euthanized, and tumors were analyzed using western blotting and immunohistochemistry.

**Acute toxicity of Z8.** Four to five-week-old female C57/BL6 mice were obtained from Vital River Laboratory Animal Technology Company Ltd and maintained in the animal husbandry facility of a specific pathogen-free (SPF) laboratory in Shengjing Hospital of China Medical University. C57/BL6 mice were used to test the acute toxicity ($LD_{50}$) of Z8 in vivo. 2500, 1000, 500, 250, and 100 mg/kg of Z8 were dissolved in normal saline containing 5% DMSO and 1% carboxymethylcellulose. Mice were randomly assigned to receive different concentrations of Z8 or vehicle control ($n = 6$ in each group). Test substance was delivered through intraperitoneal injection (i.p.), and mortality was recorded over 24 h.

**PDX models.** PDX models were established in a specific pathogen-free (SPF) laboratory facility at Shengjing Hospital affiliated with China Medical University. Human tissues were utilized according to Ethics Committee-approved protocols for research in human subjects. Three-week-old female NOD/SCID mice were obtained from Vital River Laboratory Animal Technology Company Ltd.

The tumor of TNBC patients was implanted into NOD/SCID mice and grown over three passages before harvesting for further analysis.

**Dissociation of PDX models.** Tumor fragments from the PDX model were dissociated and digested using collagenase into single-cell suspensions. Cells were cultured in Dulbecco's modified Eagle medium (DMEM) (D6429; Sigma Aldrich) supplemented with 10% fetal bovine serum and fibroblasts were excluded by digestion. After several passages (within 4), cancer cells were incubated with fluorescence-tagged antibodies and subjected to flow cytometry sorting.

**Culture of Breast cancer cell lines.** Human breast cancer cell lines (MDA-MB-231, MDA-MB-468, SKBR3, BT474, MCF7) were cultured in L15 (L1518; Sigma Aldrich), Dulbecco's Modified Eagle Medium (DMEM) (D6429; Sigma Aldrich) or 5A (M8403; Sigma Aldrich) culture medium supplemented with 10% fetal bovine serum (Hyclone). The human mammary epithelial cell line MCF10A was cultured in Mammary Epithelial Cell Medium (CM-0525, Procell, China).

Cell lines were authenticated by short tandem repeat (STR) profiling and were tested negative for *Mycoplasma*.

**Details of experimental methods**
**Fluorescence-activated cell sorting.** Tumor cells with different stem cell markers (ALDH+ cells and CD44highCD24low cells) were isolated using flow cytometry sorting (BD FACSAria III, USA). MDA-MB-231 cells and MDA-MB-468 cells were resuspended in PBS and stained with phycoerythrin (PE)-conjugated anti-CD24 at a dilution of 1:100 (Cat: 130-112-656, Miltenyi Biotec) and allophycocyanin (APC)-conjugated anti-CD44 at a dilution of 1:500 (Cat: 12211-MM02-A, SinoBiological) for 30 min at 4 °C. An ALDEFLUOR kit (Cat: # 01700, STEMCELL Technologies) was used to isolate cells with high ALDH enzymatic activity, each test tube (500 μl system) was added with 2.5 μl ALDEFLUOR reagent and each DEAB (Diethylaminoazobenzene) tube was added with 2.5 μl ALDEFLUOR reagent as well as 5 μl DEAB reagent and the

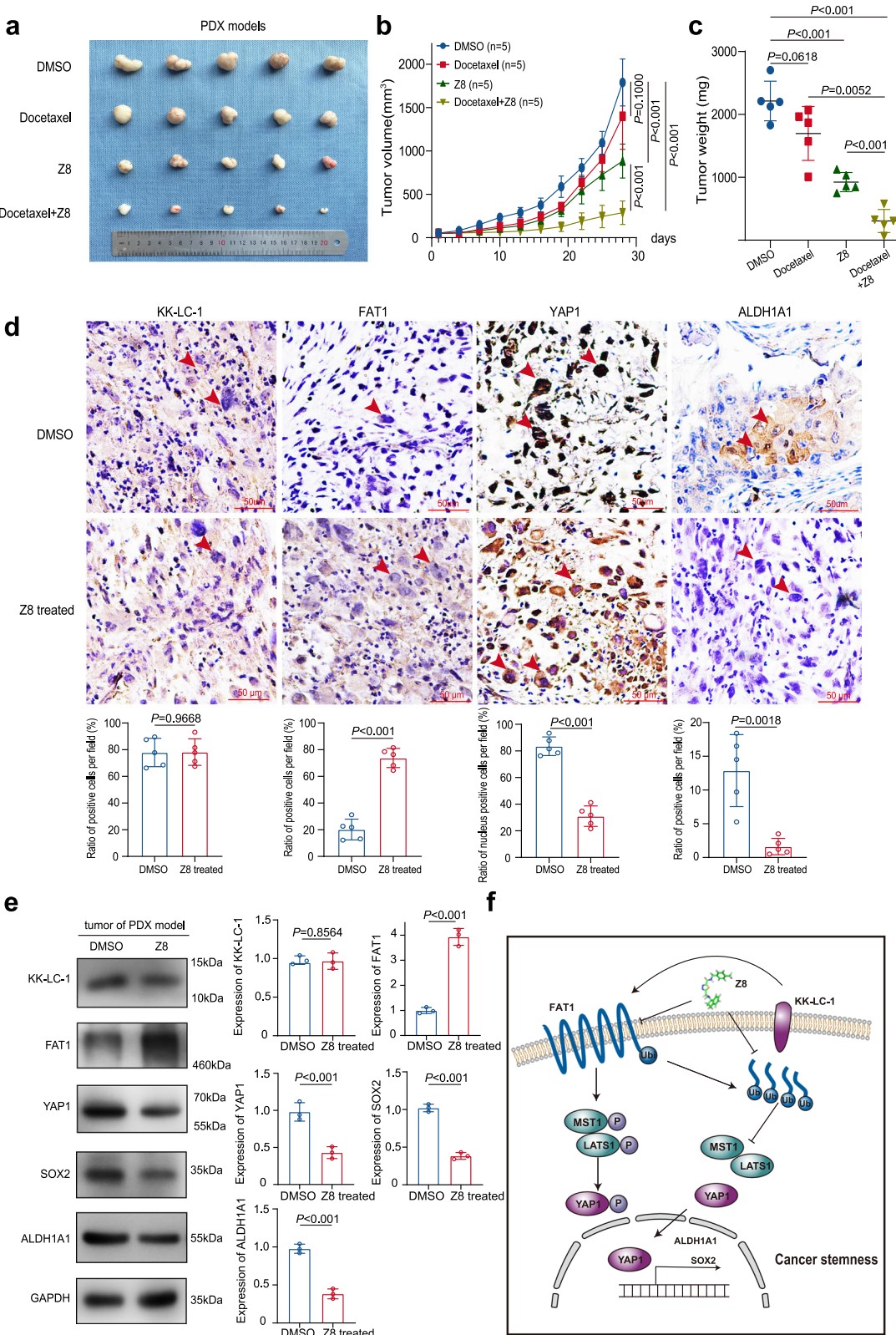

ALDH+ population was determined according to the manufacturer's instructions. The purity of the sorted populations was verified. Data were analyzed using Flowjo software (10.8.1) (BD biosciences, USA).

**CRISPR-Cas9 mediated ablation of the KK-LC-1 gene.** CRISPR-Cas9 mediated ablation of the *KK-LC-1* gene was achieved using a CRISPR-Cas9 ribonucleoprotein (RNP) complex (provided by Haixing

Bioscience) containing cassettes expressing hSpCas9 and a chimeric guide RNA. Exon 1-exon 2 of the *KK-LC-1* gene were targeted with two sgRNA sequences, AGGCGGTACTAAGTGCCGCCTGG and GCA-TATTGTAGGGTGCTCAATGG, selected from http://crispr.mit.edu. Plasmid containing the guide RNA sequence was electrotransfected into cells using the Neon transfection system, according to the manufacturer's instructions (Thermo Fisher Scientific). After two days,

**Fig. 7 | Z8 impaired tumor growth in patient-derived xenografts. a** Outcomes of PDX mouse models treated with DMSO, docetaxel, Z8, or a combination of both drugs, n = 5 mice for each group. **b** Volume of PDX in mice treated with DMSO, docetaxel, Z8, or a combination of both drugs; data are presented as mean ± SD, n = 5 mice for each group, statistical significance was determined using Student's t-test and was two-sided. **c** Weight of PDX in mice treated with DMSO, docetaxel, Z8, or a combination of both drugs; data are presented as mean ± SD, n = 5 mice for each group, statistical significance was determined using Student's t-test and was two-sided. **d** Immunohistochemistry staining showing KK-LC-1, FAT1, YAP1 and ALDH1A1 protein expression in PDX in mice treated with Z8 or DMSO, red scale bars indicate 50 μm; data are presented as mean ± SD from five randomized views, this

experiment was performed 3 times independently with similar results, statistical significance was determined using Student's t-test and was two-sided. **e:** Western blotting showing KK-LC-1, FAT1, YAP1, SOX2, and ALDH1A1 protein levels in PDX in mice treated with Z8 or DMSO; data are presented as mean ± SD from three biologically independent experiments, statistical significance was determined using Student's t-test and was two-sided. **f** Schematic depicting the mechanisms underlying KK-LC-1 regulation of ALDH⁺ cell stemness. KK-LC-1 binds FAT1 and facilitates its ubiquitination and degradation. Hippo signaling is deactivated and YAP1 is translocated to the nucleus. SOX2 and ALDH1A1 transcription is increased, promoting stemness in breast cells. Source data are provided as a Source data file.

single colonies were transferred into 96-well plates. To determine the presence of insertions or deletions (indels) in KK-LC-1 targeted clones, genomic DNA was isolated using a Quick-DNA Miniprep kit (Zymo Research), and PCR amplification was achieved using 2×Taq Master Mix (Dye Plus; Vazyme, P112) and primers in the flanking exon: Forward: 5'- TGCTGGGCTTAATTAATACCTCGG-3'; Reverse: 5'-ACAGGT-CAAATGCTGTTAGTAAGT-3'. Plasmids were isolated from 8 to 10 individual colonies and sequenced using Sanger sequencing (GENE-WIZ, China). Clones with mutations in both alleles were selected for downstream studies. All clones were maintained under the same conditions as parental cells.

**Knockdown and overexpression of the KK-LC-1 gene.** MDA-MB-231 cells and MDA-MB-468 cells were cultured in a 6-well plate and transfected with shRNAs targeting human *KK-LC-1* or NC. The sequence for sh-*KK-LC-1*#1 was 5'-GCGAAATGTCATCAAATT-3' and the target sequence for sh-*KK-LC-1*#2 was 5'- CCTCTTCTTCTGGGTTAATTA-3'. The sequence for sh-NC was 5'-GCAGTGAAAGATGTAGCCAAA-3'. For *KK-LC-1* overexpression, overexpressing plasmids were constructed by HANBIO Technologies and transfected to corresponding cells using lipo3000 (Invitrogen). The efficiency of *KK-LC-1* knockdown and overexpression was detected using qRT-PCR and western blotting.

**Knockdown of the FAT1 gene.** MDA-MB-231 cells were cultured in a six-well plate and transfected with shRNAs targeting human *FAT1* or NC. The sequence for sh-*FAT1* was 5'-CCAGGGTTTCATTCTGTCTTT-3'. The sequence for sh-NC was 5'-GCAGTGAAAGATGTAGCCAAA-3'. The efficiency of *FAT1* knockdown was detected using qRT-PCR and western blotting.

**Western blotting.** Protein synthesis was inhibited using cycloheximide at 500 nm (HY-12320; MCE) Ubiquitination-proteasomal and lysosomal degradation was inhibited with MG132 at 20 μM (HY-13259; MCE) and bafilomycin A1 at 10 nM (HY-100558; MCE), respectively. Breast tissues and cells were lysed in RIPA buffer supplemented with protease inhibitor (HY-K0010; MCE) and phosphatase inhibitor (HY-K0021; MCE). Protein concentrations in lysates were measured with a BCA Protein Assay kit (P0012S; Beyotime). Proteins were separated by electrophoresis on 6%, 10%, or 12% SDS–PAGE gels and transferred onto polyvinylidene fluoride (PVDF) membranes (Millipore). When there were great differences between molecular weights in the proteins we tested, parallel gels of the same sample were performed so that we could detect all the proteins. Non-specific binding sites on the PVDF membranes were blocked using skimmed milk for 1 h. Membranes were incubated with primary antibodies (Supplementary Table 5) and secondary antibodies, and proteins were detected using an enhanced chemiluminescence system (A680, GE).

**RNA extraction and qRT-PCR assays.** RNA in tissues and cell lines was extracted using RNAiso Plus (9108, Takara) and reverse transcribed into cDNA using a PrimeScript™ RT reagent Kit (RR037Q, Takara), according to the manufacturer's instructions. Quantitative real-time PCR (qRT-PCR) was performed on a CFX96 System (Bio-rad) using TB

Green® Premix Ex Taq™ (RR420Q, Takara). GAPDH served as the reference gene for the normalization of data. The primer sequence for each gene was provided in Supplementary Table 6.

**Co-immunoprecipitation.** Whole-cell lysates (WCL) were prepared in RIPA buffer supplemented with protease inhibitor (HY-K0010; MCE) and phosphatase inhibitor (HY-K0021; MCE). Lysates were clarified by centrifugation, incubated with primary antibodies overnight at 4 °C, and incubated with protein A/G coupled sepharose beads (L1721; Santa Cruz Biotechnology) for 2 h at 4 °C. Bound complexes were washed 3 times with RIPA buffer and eluted by boiling in SDS loading buffer. Bound proteins were detected on SDS–PAGE followed by immunoblotting.

**Protein mass spectrometry.** To detect potential proteins which may interact with KK-LC-1, we first performed the immunoprecipitation assay in MDA-MB-231 cells with anti-KK-LC-1 antibody and anti-IgG antibody according to manufacturers' instructions. When the electrophoresis process was over, the gels were cut down according to the lanes containing different samples (anti-KK-LC-1 and anti-IgG), and all molecular weights of the gels were included. Then the gels were analyzed using mass spectrometry with the help of Beijing Protein Innovation Co., Ltd, the proteins which may also be immunoprecipitated with anti-IgG antibody was removed during the analysis and a report of potential proteins which may interact with KK-LC-1 was generated.

**Immunohistochemistry analysis.** Paraffin-embedded tissue sections (4 μm thick) were deparaffinized, rehydrated, and treated with hydrogen peroxide to block endogenous peroxidase. Antigen retrieval was performed using hot citrate buffer (pH = 6) under pressure. Sections were incubated with primary antibodies (Supplementary Table 5) for 10 h, antigen-antibody binding was detected using an immunohistochemistry kit (Invitrogen), according to the manufacturer's instructions, and sections were visualized using an Olympus microscope.

**Immunofluorescence assays.** Immunofluorescence was used to localize proteins in cell lines. Briefly, cells were grown on glass coverslips, fixed in 4% paraformaldehyde for 30 min at room temperature, blocked with 5% bull serum albumin (BSA) for 1 h at room temperature, and incubated with anti-YAP1 antibody (13584-1-AP, Proteintech) at a dilution of 1:1000 overnight at 4 °C. Cells were washed in PBS, incubated with Alexa Fluor 488-(green) conjugated secondary antibodies at a dilution of 1:400 (ab150077, Abcam) for 2 h avoiding light. Cell nuclei were stained with 4',6-diamidino-2-phenylindole (DAPI) (blue). As for the immunofluorescence staining of tissue sections, the sections were deparaffinized and blocked with 5% BSA for 1 h at room temperature. Then the sections were incubated with anti-ALDH1A1 antibody (15910-1-AP, Proteintech) at a dilution of 1:500 overnight at 4 °C. The following procedures were the same as the immunofluorescence staining of cells. The results of the immunofluorescence staining were visualized using an immunofluorescence microscope (Nikon Oplenic Lumicite 9000).

**Immunofluorescence staining of tumorspheres.** ALDH+ MDA-MB-231 cells (5000 cells/ ml) with different interventions were cultured in ultra-low adhesion 6 plates (Corning) for 2 weeks to form tumor spheres. The tumorspheres were carefully transferred into 1.5 mL centrifugal tubes and centrifugated with the condition of 120 g, 5 min to remove the culture medium. The tumorspheres were carefully washed with PBS and fixed by 4% paraformaldehyde for 40 min at room temperature. Then, we dropped the tumor spheres contained in 4% paraformaldehyde onto glass slides and air-dried the paraformaldehyde. Then the tumor spheres were incubated with an anti-ALDH1A1 antibody (15910-1-AP, Proteintech) at a dilution of 1:500 or an anti-SOX2 antibody (20118-1-AP, Proteintech) at a dilution of 1:1000 overnight at 4 °C. Cells were washed in PBS, incubated with Alexa Fluor 488-(green) conjugated secondary antibody (ab150077, Abcam) at a dilution of 1:400 or Alexa Fluor 594- (red) conjugated secondary antibody (ab150080, Abcam) at a dilution of 1:400 for 2 h avoiding light. Cell nuclei were stained with 4',6-diamidino-2-phenylindole (DAPI) (blue). The tumorspheres were visualized by BioTek Cytation C10.

**Cell-counting Kit 8 (CCK-8) cell viability assay.** Cells were plated in a 96-well cell culture plate at $5 \times 10^3$ cells per well. Cells were cultured for ≤72 h in a 10% FBS culture medium. Cell viability was determined using a CCK-8 (CK04; DOJINDO) according to the manufacturer's instructions. Absorbance at 450 nm was measured using a microplate reader (Epoch; Bio-Tek Instruments, Inc). The synergistic effect of drug combination was determined by Compusyn software (ComboSyn Inc) with CI value < 1.

**Sphere formation assay.** Cells (1000 cells/ml) were cultured in ultra-low adhesion 6 plates (Corning) in serum-free Complete MammoCult™ Medium (MammoCult™ Basal Medium + MammoCult™ Proliferation Supplement + 4 μg/mL heparin + 0.48 μg/mL hydrocortisone). ALDH+ MDA-MB-231 cells and ALDH+ MDA-MB-468 cells were transduced with sh-KK-LC-1#1, sh-KK-LC-1#2 or sh-NC lentivirus to detect the effect KK-LC-1 knockdown on the sphere-forming ability of ALDH+ cells. ALDH⁻ MDA-MB-231 and ALDH⁻ MDA-MB-468 cells also interfered with the same treatment as negative controls. When analyzing the effect of Z8 on sphere formation, the small molecule drug Z839878730 (5 μM) or DMSO (1% by volume) was added to the culture medium, and tumor spheres >75 mm in diameter were counted as a sphere after 1, 2, 3, and 5 days in culture. For each analysis, five replicate wells were included in three independent experiments.

**Limiting dilution assay(LDA).** Different amounts of ALDH+ MDA-MB-231 cells and ALDH⁻ MDA-MB-231 cells (1000, 100, 10 cells/well) were cultured in ultra-low adhesion 24/96 plates (Corning) in serum-free Complete MammoCult™ Medium (MammoCult™ Basal Medium + MammoCult™ Proliferation Supplement + 4 μg/mL heparin + 0.48 μg/mL hydrocortisone). ALDH+ MDA-MB-231 cells and ALDH⁻ MDA-MB-231 cells were transduced with sh-KK-LC-1#1, sh-KK-LC-1#2, or sh-NC lentivirus to detect the effect KK-LC-1 knockdown on the self-renewing ability of ALDH+ cells, ALDH⁻ cells served as negative controls. When analyzing the effect of Z8 on the self-renewal ability of ALDH+ cells, the small molecule compound Z8 (5 μM) or DMSO (equal concentration, control) was added to the culture medium which containing ALDH+ MDA-MB-231 cells or ALDH⁻ MDA-MB-231 cells (serving as negative controls). Mammospheres >75 mm in diameter were counted and recorded after 2 weeks in culture. For each analysis, 24 replicate wells were included for the concentration of 1000 cells/well; 60 replicate wells were included for the concentration of 100, 10 cells/well. The estimated range for tumor sphere-forming cells or tumor-initiating cells were calculated using https://bioinf.wehi.edu.au/software/elda/.

**Biolayer interferometry (BLI).** The binding affinity of the small molecule to KK-LC-1 protein was determined by BLI using a Forte Bio Octet

K2 system. All assays were run at 30 °C with continuous shaking at 1000 rpm. The assay buffer consisted of PBS with 0.1% BSA, 0.01% Tween-20, and 1% DMSO. 0.15 mg/ml KK-LC-1 protein in sterile water was biotinylated and immobilized on Super Streptavidin (SSA) biosensors. Small molecule drugs were dissolved in PBS and adjusted to different concentrations. Sensors were washed with assay buffer for 10 min after each round of association and disassociation to remove nonspecifically bound protein and establish a baseline. Raw kinetic data were generated using Data Acquisition software (ForteBio). Association/dissociation rate constants ($k_{on}/k_{off}$) were generated with the double reference subtraction method using Data Analysis software (ForteBio), and affinities (KD) were calculated.

**Molecular docking of the KK-LC-1-FAT1 complex and virtual screening of small molecule compounds.** The KK-LC1 protein structure was obtained from AlphaFold Protein Structure Database (https://alphafold.ebi.ac.uk/)[31] with accession ID AF-Q5H943-F1. The query sequence of FAT1 was retrieved from UniProtKB (https://www.uniprot.org/)[32] with accession ID Q14517 and the extracellular Laminin G domain (aa: 3829–4009) was built by Modeler 10.1, with several crystal structures from Laminin G protein family as the references (PDB ID: 1C4R, 1F5F, 2JD4, 3SH4, 5CM9). The protein–protein docking was conducted by ZDOCK 3.0.2 followed by MD optimization. The virtual screening of the compound library was conducted with AutoDock Vina, and FIPSDock tools. The MM/GBSA free energy and Interfaces of the complex were analyzed by HawkDock Server (http://cadd.zju.edu.cn/hawkdock/) and PDBePISA tools (https://www.ebi.ac.uk/msd-srv/prot_int/). The 2D protein-ligand and protein–protein interaction profiles were generated using LigPlot+ and VMD educational version. Molecular dynamics (MD) simulations were performed with Gromacs 2020.3 package. All molecular graphics were displayed and prepared using the PyMOL educational version.

We first selected a number of compound libraries for virtual screening from the enamine kinase library (HY-L0061V) and the protein–protein interaction library (HY-L0066V), which consist of ~70,000 small molecules. Molecular dynamics simulations (MD) were performed to optimize the complex structure of KK-LC-1 and FAT1. The key amino acid residues at the complex binding interface were evaluated to define the virtual screening pocket (center_x = 36.294; center_y = 49.379; center_z = 37.191). The compounds were then evaluated and ranked according to binding free energies. The top 10% of small molecules were filtered by druggability properties to remove molecules with potential toxicity or poor drug-like abilities and the small molecules we selected were then subjected to bio-layer interferometry.

**Statistical analysis.** Statistical analyses were performed using IBM SPSS v22 (SPSS, Armonk, NY, USA) and GraphPad Prism version 7. Descriptive statistics except for drug susceptibility tests are reported as means ± standard deviations (SDs). Drug susceptibility tests are displayed as means ± standard error of the mean (SEM). Between-group differences were assessed using chi-squared or Student's t-tests. The Kaplan–Meier method was used for survival analyses and P values were generated by log-rank test. Results were considered significant at $P < 0.05$.

## Data availability

The TCGA publicly available data used in this study are available via (https://portal.gdc.cancer.gov/)[19], molecular and clinical data for the TNBC samples were obtained from a previous publication[20]. The association of KK-LC-1 mRNA expression and clinical outcomes could be achieved via (http://kmplot.com)[21]. The mRNA seq data generated in this study have been deposited in the GSA database under accession code HRA003964 and HRA003962. Kyoto Encyclopedia of Genes and Genomes (KEGG) pathway enrichment data could be obtained via

(https://www.kegg.jp/kegg/pathway.html)[25]. The mRNA expression data of *KK-LC-1* in various breast cancer cell lines were obtained from Cancer Cell Line Encyclopedia (CCLE) (https://sites.broadinstitute.org/ccle/tools)[22]. The mass spectrometry proteomics data are deposited in the ProteomeXchange with the identifier PXD040200. The processed data are available in OMIX database (OMIX001925, OMIX001926, and OMIX002950). The KK-LC1 protein structure was obtained from AlphaFold Protein Structure Database (https://alphafold.ebi.ac.uk/)[31] with accession ID AF-Q5H943-F1. The query sequence of FAT1 was retrieved from UniProtKB (https://www.uniprot.org/)[32] with accession ID Q14517. The remaining data are available within the Article, Supplementary Information, or Source Data file. Source data are provided with this paper and are also available in figshare at: https://doi.org/10.6084/m9.figshare.22598071.

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

## Acknowledgements

This study was supported by the National Natural Science Foundation of China (Grant: U20A20381, 81872159, 82203804); Y.-L.Y.'s laboratory was supported by the National Natural Science Foundation of China (Grant: 81874301), the Fundamental Research Funds for Central University (Grant: DUT22YG122) and the Key Research project of 'be Recruited and be in Command' in Liaoning Province (2021JH1/10400050).

## Author contributions

C.-G.L. and Y.-L.Y. designed the project and wrote the manuscript. J.-W.B., S.-J.W., Y.-X.Z., L.-S.S., X.-D.Z., and Q.-T.M. performed most of the experiments including mice xenograft models, breast tumor cell isolation, flow cytometry staining, in vitro cell culture, western blotting, qRT-PCR, and original data analysis. J.-W.B., S.-J.W., and L.-S.S. performed with flow cytometry analysis. X.-D.Z. and Q.-T.M. performed the isolation of primary cancer cells. H.-N.L. and Y.-L.Y. helped with small molecule screening and molecular docking. Y.L. helped with the chemical synthesis of the small molecule compound. C.L. and C.-G.L. performed bioinformatics analysis. G.-L.C., N.N., and J.-Q.X. helped with the collection of clinical samples. L.-S.S., Y.-L.Y., and C.-G.L. supervised this study. L.-S.S., H.-N.L., Y.-X.Z., Y.-L.Y., and C.-G.L. helped with manuscript editing. C.-G.L., L.-S.S., and Y.-L.Y. provided the project funding.

## Competing interests

The authors declare no competing interests.
