## [Peer Review File · Nature Communications]

Reviewers' Comments:

Reviewer #1:

Remarks to the Author:

Author: Bu et al.

Title: KK-LC-1 represents a feasible therapeutic target to eliminate ALDH+ stem cells in triple negative breast cancer

Comments:

A. Summary of the key results:

In this manuscript, Bu et al provide the first evidence that KK-LC-1-FAT1-Hippo-ALDH1A1 signaling axis plays important roles in ALDH+ TNBC stemness and tumorigenesis. They demonstrate both in vitro and in vivo that targeting KK-LC-1-FAT1 protein-protein inaction (PPI) is a novel therapeutic strategy for the treatment of ALDH+ TNBC.

B. Validity:

There is no major flaw which prohibit its publication.

C. Originality and significance:

Although many small molecules targeting the components of the Hippo pathway have been reported (Cunningham and Hansen, 2022; Dey et al., 2020; Wu et al, 2018), this is the first report on targeting the KK-LC-1-FAT1 for TNBC therapy. The research findings provide strong preclinical data for future clinical trials for the treatment of TNBC and drug resistant BC in the future.

D. E. Data & methodology & appropriate use of statistic and treatment of uncertainty

OK

F. Conclusion:

The major conclusion is mostly supported by the experimental findings. However, it is premature to conclude from the data that targeting KK-LC-1-FAT1 PPI only inhibits ALDH+ TNBC. No data has been provided to show that targeting KK-LC-1-FAT1 has no effect on ALDH- TNBC. This manuscript will not be considered to be accepted by Nat Commun after the major revisions suggested below.

G. Suggested improvement

Major

1. There is no list of genes upregulated in TNBC and DTX resistant BC cells (Fig. 1b). It is unclear why only 10 genes were found upregulated in TNBC cancer.

2. KK-LC-1 issues:

1) CRISPR is used to knock out KK-LC-1 in TNBC cells (Extended Fig. 1h). However, there is no western blot to show that KK-LC-1 is indeed knocked out.

2) To eliminate off-target effect of CRISPR or shRNAs, at least two sgRNAs or shRNAs are required for at least major experiments.

3) CRISPR KK-LC-1 knockout in TNBC cells significantly increases ALDH+ cells (Fig. 1f-h). It is well-known that shRNAs have much higher off-target effects than CRISPR. However, it is unclear that instead of using the CRISPR KK-LC-1 KO lines, shRNA-mediated KK-LC-1 knock down lines were used in the rest of the experiments.

3. It is unclear from the manuscript how KK-LC-1 regulates FAT1 stability.

4. Z8 should be also tested in KK-LC-1-/- (CRISPR KO) TNBC lines to support its specificity.

5. For all ALDH+/NC and ALDH+/sh-KK-LC-1 experiments (Fig. 2) or Z8 drug treatments (Fig. 6), ALDH- TNBC cells should be used as negative controls to support the conclusion that ALDH+ cells have higher tumorigenic potential and inhibition of KK-LC1-FAT1 PPI specifically inhibit ALDH+ cells.

6. Since KK-LC-1 is upregulated in DTX-resistant TNBC cells (Fig. 1b), Z8 should be used to test

whether it can also reduce DTX drug resistance in these cells.

7. Hippo-independent effect of KK-LC-1-FAT1 on TNBC should at least be discussed.

Minor

1. Extended Data Fig. 2c: the quality of the WB for KK-LC-1 is poor. Knockdown of KK-LC-1 by sh-KK-LC-1 is not very convincing.

2. Transcriptional up-regulation of ALDH1A1 by TAZ in cancer stem cells has already been reported previously (doi: 10.18632/oncotarget.16430), which should be cited.

3. "transfected with lentivirus" should be "transduced with lentivirus" or "infected with lentivirus".

4. Page 21, line 599: "CT83" should be "KK-LC-1".

H. Reference: OK

I. Clarity and context: OK

Reviewer #2:

Remarks to the Author:

The manuscript by Bu et al reported that KK-LC-1, a cancer testis antigen, dictates the stemness of ALDH+ TNBC cells via binding to FAT1 and promotes its ubiquitination and degradation. Moreover, they screened small-molecule compounds that can disrupt the binding interface between KK-LC-1 and FAT1 via computational approaches and demonstrated that a small-molecule compound Z8 can suppress TNBC tumor growth via reactivating the Hippo pathway and reducing the stemness and viability of ALDH+ TNBC cells both in vitro and in vivo. Indeed, KK-LC-1 has been deemed as a potential biomarker for various solid tumors in previous studies. Yet, the detailed molecular mechanisms behind the role of KK-LC-1 in the pathogenesis of solid tumors still remain elusive. This study revealed that the KK-LC-1-FAT1-Hippo-YAP-ALDH1A1 signaling axis might be a novel target to reverse the ALDH+ TNBC cell stemness and viability through a series of experiments. Followings are some concerns that need to be addressed.

1. In Figure 1, KK-LC-1 was revealed through the comparison of docetaxel-resistant MDA-MB-231 cells and its' parental cell line. However, there is no evidence to confirm the drug resistance phenotype of the drug-resistant cell line. The author should provide at least the drug susceptibility data of these two cell lines.

2. It seems that the authors have sufficient clinical samples to explore the relevance of the prognosis of TNBC patients and KK-LC-1 expression. I suggest that the data of overall survival and disease-free survival of the patients involved in Figure 3j should be included.

3. In Figure 4, the author tried to demonstrate that KK-LC-1 knockout could affect the Hippo signaling pathway (MST1, pMST1, LATS1, pLATS1, YAP1 and pYAP1) via FAT1. To assure that FAT1 serves as the link between KK-LC-1 and the Hippo signaling pathway, I suggest that a rescue experiment should be conducted. The author should knock down FAT1 in the KK-LC-1-/- MDA-MB-231/468 cells to see if the Hippo signaling pathway could be deactivated.

4. Indeed, a pile of previous studies have been geared towards ALDH+ TNBC stem-like cells. The authors should elaborate in a paragraph to discuss and comment the uniqueness and limitation of their discovery in the Discussion Section.

5. More details are needed about the computational discovery of small-molecule inhibitor that may disrupt the KK-LC-1 and FAT1 binding interface.

6. What are the advantages of using computational approach for the discovery of protein-protein interface inhibitors in this work? The authors should elaborate in the Discussion Section.

7. Some of the fonts in the figures are too small.

8. The band quality of KK-LC-1 in Figure 3f and Figure 3h should be improved.

9. The molecular weights of the proteins for WB should be labeled in the figures.

10. The author should indicate the exact number of mice used in the in vivo drug susceptibility in the figures and the figure legends.

Reviewer #3:

Remarks to the Author:

The current work describes the identification of a KK-LC-1-FAT1-Hippo-YAP-ALDH1A1 pathway in breast cancer that would be important for controlling the stemness of the cellular subpopulations in the tumor and therefore its resistance capacities: in some triple negative breast cancer cells KK-LC-1 is overexpressed and the authors demonstrate robustly that this leads indirectly to increased ALDH1 activity, resulting in highly tumorigenic cancer stem cells. In addition, KK-LC-1 is validated as target for designing novel drugs with capacity for overcoming resistance by hampering stem cell formation in the tumor of breast cancer.

I think this is a quite sound work, with a robust experimental approach with multiple complementary experimental techniques that seem to fit and solidly demonstrate the validity of the proposed target for anticancer drug design. However, I still see some points that would require improvement before being ready for publication. Namely:

* The description of the computational approach is really poor. Since both the KK-LC-1 and FAT1/Laminin G domain are both modeled (there are no experimental structures for them), how was the protein-protein interface identified? How was the binding pocket used for the virtual screening of compounds selected? Did it involve only KK-LC-1 or both KK-LC-1/FAT1? What compound collection was used? The description of the methods used seem rather a black box by just mentioning the computational tools used.

* In Figure 1e, ALDH⁻ cells show near 10K-fold higher expression of KK-LC-1 than ALDH⁺ cells, while in the text (eg. line 138) the opposite message is conveyed (ALDH⁺ cells have higher KK-LC-1 expression). Is this a bug in the Figure labels?

* Please provide explanation for "CTA" in lines 82 and 283.

* Minor issues: multiple typos in the text (eg "resisitance", "resisant" instead of resistance and resistant in page 5 multiple times, "populations" instead of populations in the same page, etc)

Reviewer #4:

Remarks to the Author:

The authors of this manuscript describe a novel signalling axis in TNBC that promotes breast cancer stemness and is associated with therapy resistance. The authors' main findings are that 1) KK-LC-1 is highly expressed in ALDH-positive TNBC tumours and cell lines and is associated with cancer drug resistance, stemness, tumorigenicity, and poor survival; 2) loss of KK-LC-1 results in loss of tumorigenicity, decreased tumour initiation and tumour growth rates; 3) KK-CL-1 directly interacts with FAT1 to promote its degradation via the 26S ubiquitin-proteasome pathway; 4) which results in the dysregulation of the Hippo pathway and increased production and nuclear localisation of YAP transcription factor. Through their findings, the authors have identified a novel therapeutically relevant pathway that is validated in patient-derived tissues and identified a promising therapeutic molecule, Z8, targeting this pathway.

General comments:

- In several instances, the authors include identical data in both the main figures and the extended figures, duplicates should be removed to avoid confusion.
- The authors should indicate the sample sizes for animal experiments and tumour samples in the figure legends and methodology in cases where it is not stated.
- The authors appear to interchange between using HSP90 and GAPDH as loading controls/house-keeping proteins, they should explain why the loading control is not kept consistent throughout the manuscript.
- A positive control hippo pathway activator is an important control that could have been included throughout the manuscript and the authors should address why a positive control was not included.

- Nature Communications endorses the recommendations of the ARRIVE guidelines for animal experiments and therefore the authors should include more detailed descriptions of the animal husbandry and housing, randomisation and blinding, sample sizes, and rationale for strains chosen etc when describing their animal experiments. Importantly, the animal ethics authorization reference numbers should be quoted which are missing from this manuscript.
- There are some spelling and grammatical errors throughout the manuscript, particularly on page 5 and in the discussion section, that if fixed would improve the manuscript's readability.

Major comments

- page 5, lines 138-140: The authors describe KK-LC-1 to be overexpressed in the ALDH+ population, however, the graph in figure 1e shows the exact opposite result. This is likely a simple error but needs to be addressed. It should also be described/mentioned in the results or figure legend, how this data was generated.
- On page 5, lines 143-145, the authors describe KK-LC-1 expression as highly tumor specific. Do the authors mean specific to TNBC tumours or is this a pan-cancer phenomenon? This is not clear. If it is the latter then this contradicts the data in figure 1B and needs explaining.
- From figure 1F onwards, why were the KK-LC-1 knockdown experiments not carried out in the docetaxel-resistant cell line, given that the data in 1a indicates KK-LC-1 overexpression to be specific to these cells compared to parental cells?
- The mass-spectrometry description for figure 3b is missing from the materials and methods and needs to be included with a description of the machinery used and the sample preparation steps.
- The comparisons for figure 3h are made across two separate gels, ideally, the protein should have been run on one gel. Were these gels processed in parallel? This information should be disclosed in the manuscript.
- the conclusion drawn from the data in figures 4e-f is weak and cannot be substantiated without proper quantification of the immunocytochemistry results.

Minor comments

- In the introduction "CTA" should be written in full before use of the acronym.
- On page 5, line 126, the authors mention "upregulated genes in TNBC." The database used to acquire this list of genes needs to be mentioned somewhere.
- The methods for figure 2d do not seem to be described in the materials and methods in terms of how spheroids were processed for immunofluorescence.
- A quantification for figure 2e would be a useful addition with statistical significance displayed.
- As per Nature Communication guidelines, uncropped, unprocessed gels should be made available and this will be relevant for figure 3h FAT1 blots where there appear to be additional higher weight bands cropped out of the image that might be relevant to reviewers and the eventual readers of this manuscript.

Dear Editor,

We are very pleased to have been given the precious opportunity to revise our manuscript for ***Nature Communications***. We sincerely want to thank you and the referee's insightful comments and suggestions. Accordingly, we have conducted additional experiments and analysis to address the referee's concerns and improved the quality of our work. We would like to extend our appreciation to you for the valuable time and kind efforts to provide guidance for the manuscript. We amended the manuscript per each suggestion from the reviewers and responded in point-to-point manner with a clear indication of the revised location. We hope these will make the manuscript more acceptable for publication.

Responses to Referee #1's Comments:

General Comments: ***".....In this manuscript, Bu et al provide the first evidence that KK-LC-1-FAT1-Hippo-ALDH1A1 signaling axis plays important roles in ALDH+ TNBC stemness and tumorigenesis. They demonstrate both in vitro and in vivo that targeting KK-LC-1-FAT1 protein-protein inaction (PPI). Although many small molecules targeting the components of the Hippo pathway have been reported (Cunningham and Hansen, 2022; Dey et al., 2020; Wu et al, 2018), this is the first report on targeting the KK-LC-1-FAT1 for TNBC therapy. The research findings provide strong preclinical data for future clinical trials for the treatment of TNBC and drug resistant BC in the future."***

RESPONSE: Many thanks for your very kind words about our work. We are excited to hear that you think our manuscript is relatively novel with translational impact to the field of protein-protein inhibitors (PPI) and TNBC therapy. We are grateful for your valuable time and kind efforts you have spent to provide your very insightful comments, which are of great help for our revision.

Comment 1: ***“There is no list of genes upregulated in TNBC and DTX resistant BC cells (Fig. 1b). It is unclear why only 10 genes were found upregulated in TNBC cancer.”***

RESPONSE: We want to thank the referee for this very helpful comment. This has been fixed. Indeed, there were 56 genes upregulated in TNBC cancerous tissue compared to the normal tissue in TCGA database and 520 genes upregulated in DTX resistant MDA-MB-231 cells. Per the referee’s comment, we have carefully labeled the upregulated genes in Figure 1b as below. We confirmed that there exists two overlapped genes between the two groups, KK-LC-1 and MMP1. Please see **Figure 1b** and the detailed gene lists in the **Related Manuscript file 1** for more details.

Figure 1b

Comment 2: ***“CRISPR is used to knock out KK-LC-1 in TNBC cells (Extended Fig. 1h). However, there is no western blot to show that KK-LC-1 is indeed knocked out.”***

RESPONSE: Many thanks for this critical comment. This has been fixed. We have conducted additional Western experiments to address the concern. The expression of KK-LC-1 in KK-LC-1+/+ MDA-MB-231 cells and KK-LC-1-/- MDA-MB-231 cells were revealed in **Figure 3e** as below. Moreover, the expression of KK-LC-1 in KK-LC-1+/+ MDA-MB-468 cells and KK-LC-1-/- MDA-MB-468 cells also were detected by Western blot and revealed in **Supplementary Figure 3a** as below. Please see draft-track-change for more details.

Figure 3e

Supplementary Figure 3a

Comment 3: ***“To eliminate off-target effect of CRISPR or shRNAs, at least two sgRNAs or shRNAs are required for at least major experiments.”***

RESPONSE: We appreciate for this critical and very helpful comment. This has been fixed. Per the referee’s suggestion, we have reconducted the sphere formation assay by two shRNAs knocking down KK-LC-1 as seen in **Figure 2a-2d** below with ALDH- MDA-MB-231/468 cells as negative controls. Moreover, we have reconducted the immunofluorescence staining of ALDH1 using 2 shRNAs as shown in **Figure 2e**. The efficacy of the shRNAs for KK-LC-1 was detected using qRT-PCR and Western blot in **Supplementary Figure 2c and 2d**. Furthermore, we have reconducted the limiting dilution assay using 2 shRNAs of KK-LC-1 as in **Supplementary Table 1**.

Fig. 2a-d

Fig. 2e

Supplementary Figure 2c and 2d

As for the concern of CRISPR off-target effects, indeed, we initially used two sgRNAs to knock out KK-LC-1 to prevent off-target effect. Specifically, these two sgRNAs were designed to target the region between Exon 1~Exon 2 of the KK-LC-1 gene to obtain genetically deletion of KK-LC-1. This strategy was believed to largely avoid off-target effects. At a result, only one colony was produced using two sgRNAs. To avoid biased results due to off-target effects, we also supplemented two additional shRNAs to knock down KK-LC-1 for partial experiments in differential TNBC cell lines to reveal the functional and molecular mechanism that regulated by KK-LC-1.

A scheme of the CRISPR knockout strategy for KK-LC-1 is provided as below:

The sequences of the two sgRNAs are provided as below:

- ① sgRNA-A1: AGGCGGTACTAAGTGCCGCC TGG
- ② sgRNA-A2: GCATATTGTAGGGTGCTCAA TGG

Comment 4: “**CRISPR KK-LC-1 knockout in TNBC cells significantly increases ALDH+ cells (Fig. 1f-h). It is well-known that shRNAs have much higher off-target effects than CRISPR. However, it is unclear that instead of using the CRISPR KK-LC-1 KO lines, shRNA-mediated KK-LC-1 knock down lines were used in the rest of the experiments.**”

RESPONSE: We are grateful for this critical and insightful comment. We fully agree with the referee that shRNAs may have higher off-target effects than CRISPR. Nevertheless, more ALDH+ cells are required in our experiments and therefore it was necessary to avoid ALDH+ cells from undergoing differentiation in culture for as long as possible. Hence, we decided to adopt the approach by Zhao et al.¹ on ALDH+ cells using shRNAs for the knockdown of KK-LC-1.

Below are specific details for the experiments,

In brief, we first sorted out ALDH+ cells from MDA-MB-231/468 cells using flow cytometry and then interfered with KK-LC-1 expression using shRNAs. Notably, if we sorted out ALDH+ from KK-LC-1-/- MDA-MB-231/468 cells, the proportion of ALDH+ cells was too low to efficiently sort out for subsequent experiments, and it would take longer to expand even the small number of ALDH+ cells, and the probability of differentiation would be higher the longer the culture time. Second, after sorting out ALDH+ cells from wild-type MDA-MB-231/468 cells, these ALDH+ cells may differentiate during prolonged in vitro culture, and these ALDH+ cells will take even longer if they are then knocked out using CRISPR until a monoclonal strain is screened. In order to avoid biased results, we finally used shRNAs, rather than CRISPR, to interfere with the expression of KK-LC-1 in ALDH+ cells.

REFERENCE:

1 Zhao, L. et al. SGCE Promotes Breast Cancer Stem Cells by Stabilizing EGFR. *Adv Sci (Weinh)* 7, 1903700 (2020).

Comment 5: ***“It is unclear from the manuscript how KK-LC-1 regulates FAT1 stability.”***

RESPONSE: Many thanks for this critical comment. To clarify this concern, we have elaborated in a paragraph in the manuscript to explain how KK-LC-1 regulates the stability of FAT1. Please see below or draft-track-change for more details,

“We found that KK-LC-1 could bind with FAT1 in MDA-MB-231/468 cells and their expression were negatively correlated (Fig. 3 c-f, j-k, Fig. 3 a). Further on, we found that the existence of KK-LC-1 could significantly accelerate the degradation of FAT1, which led to the relative low expression of FAT1. Then, we tested the ubiquitination of FAT1 in conditions with or without KK-LC-1 and found that KK-LC-1 could increase the ubiquitination of FAT1 thus making FAT1 more easily to be degraded by ubiquitin-proteasomal pathway. Overall, we

found that *KK-LC-1* could bind with *FAT1* and increase the ubiquitination of *FAT1* which led to the instability of *FAT1*(degrading more quickly).”

Comment 6: “**Z8 should be also tested in *KK-LC-1*^{-/-} (CRISPR KO) TNBC lines to support its specificity.**”

RESPONSE: We want to thank the referee for this expert comment and suggestion. Per the referee’s suggestion, we have conducted drug susceptibility test of Z8 on *KK-LC-1*^{-/-} MDA-MB-231 and MDA-MB-468 cells. Interestingly, we found that cell viability of two cell lines was barely suppressed upon the treatment of Z8, implicating the potential selectivity of Z8 compound. Please see **Supplementary Figure 9b** as below or **draft-track-change** for more details.

Supplementary Figure 9b

Comment 7: “**For all *ALDH*⁺/*NC* and *ALDH*⁺/*sh-KK-LC-1* experiments (Fig. 2) or Z8 drug treatments (Fig. 6), *ALDH*⁻ TNBC cells should be used as negative controls to support the conclusion that *ALDH*⁺ cells have higher tumorigenic potential and inhibition of *KK-LC1-FAT1* PPI specifically inhibit *ALDH*⁺ cells.**”

RESPONSE: Many thanks for this very helpful comment. Per the referee’s suggestion, we have reconducted the sphere formation assay (**Figure 2a-d** and **Figure 6c-d**), using *ALDH*⁻ TNBC cells as negative control. Our findings revealed that *ALDH*⁺ cells exhibit stronger sphere formation ability than *ALDH*⁻ cells. Moreover, the inhibition of *KK-LC-1-FAT1* PPI can specifically inhibit *ALDH*⁺ cells. We hope these extra experiments could clarify the concerns.

Figure 2a-d

Figure 6c-d

Comment 8: “**Since *KK-LC-1* is upregulated in DTX-resistant TNBC cells (Fig. 1b), Z8 should be used to test whether it can also reduce DTX drug resistance in these cells.**”

RESPONSE: We are truly grateful for this expert comment and suggestion. Per the referee's suggestion, we have tested the effects of Z8 on DTX-resistant MDA-MB-231 cells. Interestingly, we found that Z8 could reduce the DTX drug resistance in these cells (**Supplementary Figure 9c**). Furthermore, we examined the function of KK-LC-1 in DTX-resistant MDA-MB-231 cells and we found that the ratio of ALDH⁺ cells were significantly reduced after KK-LC-1 knocking down (**Supplementary Figure 2f-g**).

Supplementary Figure 9c

Supplementary Figure 2f-g

Comment 9: ***“Hippo-independent effect of KK-LC-1-FAT1 on TNBC should at least be discussed.”***

RESPONSE: Many thanks for this critical comment. Per the referee's suggestion, we have elaborated the Hippo-independent effect of KK-LC-1-FAT1 in TNBC cells in the discussion section of the manuscript. Please see draft-track-change for more details (line 486-496).

Minor comments

Comment 9: “**Extended Data Fig. 2c: the quality of the WB for KK-LC-1 is poor. Knockdown of KK-LC-1 by sh-KK-LC-1 is not very convincing.**”

RESPONSE: This has been fixed. To clarify this concern, we have replaced the WB band in **Supplementary Figure 2c** as below.

Supplementary Figure 2c

Comment 10: “**Transcriptional up-regulation of ALDH1A1 by TAZ in cancer stem cells has already been reported previously (doi: 10.18632/oncotarget.16430), which should be cited.**”

RESPONSE: Many thanks for this helpful suggestion. This has been fixed. Per the referee’s suggestion, we have cited this paper² regarding the transcriptional up-regulation of ALDH1A1 by TAZ in cancer stem cells.

REFERENCE:

2. Yu J, et al. TAZ induces lung cancer stem cell properties and tumorigenesis by up-regulating ALDH1A1. *Oncotarget* 8, 38426-38443 (2017).

Comment 11: “**transfected with lentivirus**” should be “**transduced with lentivirus**” or “**infected with lentivirus**”.

RESPONSE: This has been fixed. Per the referee’s comment, we have amended the phrase as “transduced with lentivirus”.

Comment 12: “**Page 21, line 599: “CT83” should be “KK-LC-1”.**”

RESPONSE: This has been fixed. Per the referee's comment, we have amended "CT83" as "KK-LC-1" in line 599. Please see draft-track-change for more details.

CLOSING RESPONSE TO REFEREE #1:

We are truly grateful for all of your insightful comments. Again, thank you for taking the precious time and kind efforts to help us improve our work.

Reviewer #2 - ALDH1+ BCSCs -(Remarks to the Author):

General Comments: “.....***This study revealed that the KK-LC-1-FAT1-Hippo-YAP-ALDH1A1 signaling axis might be a novel target to reverse the ALDH+ TNBC cell stemness and viability through a series of experiments. Followings are some concerns that need to be addressed.***”

RESPONSE: Many thanks for your kind words about our work. We are grateful that you think our manuscript is relatively novel. We truly appreciate for your valuable time and kind efforts you have spent to provide your very insightful comments, which are of great help for our revision.

Comment 1: “***In Figure 1, KK-LC-1 was revealed through the comparison of docetaxel-resistant MDA-MB-231 cells and its' parental cell line. However, there is no evidence to confirm the drug resistance phenotype of the drug-resistant cell line. The author should provide at least the drug susceptibility data of these two cell lines.***”

RESPONSE: Thank you for this very insightful comment. Per the referee's comment, we have conducted extra experiments to test the drug susceptibility on MDA-MB-231 parental cells and docetaxel-resistant MDA-MB-231 cells to reveal their phenotype difference (**Supplementary Figure 1a as below**).

Supplementary Figure 1a

Comment 2: ***“It seems that the authors have sufficient clinical samples to explore the relevance of the prognosis of TNBC patients and KK-LC-1 expression. I suggest that the data of overall survival and disease-free survival of the patients involved in Figure 3j should be included.”***

RESPONSE: Many thanks for this critical comment. Per the referee’s suggestion, we have conducted the survival analysis on TNBC patients upon KK-LC-1 expression (**Supplementary Figure 1f**). The results showed that KK-LC-1 was significantly associated with worse disease-free survival and overall survival in TNBC patients.

Supplementary Figure 1f

Comment 3: ***“In Figure 4, the author tried to demonstrate that KK-LC-1 knockout could affect the Hippo signaling pathway (MST1, pMST1, LATS1, pLATS1, YAP1 and pYAP1) via FAT1. To assure that FAT1 serves as the link between KK-LC-1 and the Hippo signaling pathway, I suggest that a rescue experiment should be conducted. The author should knock down FAT1 in the KK-LC-1^{-/-} MDA-MB-231/468 cells to see if the Hippo signaling pathway could be deactivated.”***

RESPONSE: We are grateful for this expert comment and suggestion. Per the referee’s suggestion, we have conducted extra experiments to clarify the concerns. First, we examined the efficacy of the shRNA for FAT1 using qRT-PCR and Western blot in **Supplementary Figure 4a-b**. Subsequently, we have conducted the rescue experiment in the Hippo signaling pathway (MST1, pMST1, LATS1, pLATS1, YAP1 and pYAP1) upon FAT1 knock down in the KK-

LC-1^{-/-} MDA-MB-231/468 cells. Interestingly, we found that the knock down of FAT1 can significantly deactivate Hippo signaling pathway (**Figure 4a-b**).

Supplementary Figure 4a-b

Fig. 4a-b

Comment 4: ***“Indeed, a pile of previous studies have been geared towards ALDH+ TNBC stem-like cells. The authors should elaborate in a paragraph to discuss and comment the uniqueness and limitation of their discovery in the Discussion Section.”***

RESPONSE: Many thanks for this critical comment. Per the referee’s comment, we have elaborated a paragraph to comment the uniqueness and limitation of

the previous studies about ALDH+ TNBC stem-like cells in the Discussion section. Please see draft-track-change for more details (line 418-431).

Comment 5: “**More details are needed about the computational discovery of small-molecule inhibitor that may disrupt the KK-LC-1 and FAT1 binding interface.**”

RESPONSE: We want to thank the referee for this very helpful comment. This has been fixed. Per the referee’s comment, we have added more details to carefully describe the computational discovery of small-molecule inhibitor that may disrupt the KK-LC-1 and FAT1 binding interface. Please see draft-track-change (line 771-794; line 324-336) or below for more details.

The KKLC1 protein structure was obtained from AlphaFold Protein Structure Database (<https://alphafold.ebi.ac.uk/>) with accession ID AF-Q5H943-F1. The query sequence of FAT1 was retrieved from UniProtKB with accession ID Q14517 and the extracellular Laminin G domain (aa: 3829-4009) was built by Modeller 10.1, with several crystal structures from Laminin G protein family as the references (PDB ID: 1C4R, 1F5F, 2JD4, 3SH4, 5CM9). The protein-protein docking was conducted by ZDOCK 3.0.2 followed by MD optimization. The virtual screening of compound library was conducted with AutoDock Vina, FIPSDock tools³. The MM/GBSA free energy and Interfaces of complex were analyzed by HawkDock Server (<http://cadd.zju.edu.cn/hawkdock/>) and PDBePISA tools (https://www.ebi.ac.uk/msd-srv/prot_int/). The 2D protein-ligand and protein-protein interaction profiles were generated using LigPlot+ and VMD educational version. Molecular dynamics (MD) simulations were performed with Gromacs 2020.3 package. All molecular graphics were displayed and prepared using the PyMOL educational version.

We first selected a number of compound libraries for virtual screening from the enamine kinase library (HY-L0061V) and the protein-protein interaction library (HY-L0066V), which consist of ~70,000 small molecules. Molecular dynamics simulations (MD) were performed to optimize the complex structure of KK-LC-1 and FAT1. The key amino acid residues at the complex binding interface were evaluated to define the virtual screening pocket (center_x = 36.294; center_y =

49.379; center_z = 37.191). The compounds were then evaluated and ranked according to binding free energies. The top 10% of small molecules were filtered by druggability properties to remove molecules with potential toxicity or poor drug-like abilities. Finally, 145 compounds were evaluated via the bio-layer interferometry (BLI) experimental assay for binding affinity. Among them, Z839878730 (Z8) was found to have the strongest binding ability and was predicted to disrupt the binding KK-LC-1 to FAT1. (Fig. 5d-f and SI Fig.1).

SI Fig. 1. The snug-fit-in model of Z839878730-KK-LC1-Laminin G complex in the black circle box. The green dash represents the putative hydrogen binding; The arcs represent the putative hydrophobic interactions.

“After 100 ns MD simulations, we found that the interaction between KK-LC-1 and Laminin G domain was significantly disrupted in the complex system (Fig. 5d-f). The small molecule occupied the groove composed by Sheet-1, Sheet-12 and several loop structures. Hence, the C-terminal part of KK-LC1 helix structure become difficult to interact with Laminin G domain, which contributes the most favorable interaction between these two proteins. The helical structure was even breakdown at the latter part of the trajectory in the complex system during the 100ns MD simulation.

In particular, to assess the putative capability of small-molecule inhibitor disrupting the KK-LC-1 and FAT1 binding interface, we further performed the Steered Molecular Dynamics (SMD) simulations on the equilibrated model after the 100ns MD simulations. A constant pulling velocity was applied to KK-LC-1

to pull it away from the binding surface of the Laminin G domain. To achieve this, $50 \text{ kJ mol}^{-1} \text{ nm}^{-2}$ soft elastic spring and 0.001 nm ns^{-1} rupture force were applied to KK-LC-1 along the pulling direction, whereas the Laminin G domain was chosen as the static reference throughout the pulling process. Snapshots of the structure were captured every 200 ps and steering forces were recorded every 50 ps. The models were pulled for 1 ns. During the 1 ns SMD simulations (Supplementary Fig. 8), we found that KK-LC-1 can easily depart from FAT1 in Z839878730-KK-LC1-Laminin G complex with lower force as compared to KK-LC1-FAT1 complex. The distances between the center of mass of the KK-LC-1 and that of FAT1 before and after pulling were 1.953 nm and 2.953 nm in KK-LC1-FAT1 complex, and were 1.259 nm and 2.259 nm in Z839878730-KK-LC1-Laminin G complex system, respectively. The pulling direction for KK-LC-1 was chosen to achieve effective separation and all of the interactions between the KK-LC-1 and FAT1 disappeared around 400ps. These results implicated that Z8 may be able to disrupt the binding between KK-LC-1 to FAT1.”

Supplementary Figure 8.

a-b: Structure comparisons of KK-LC1-FAT1 system during the 1 ns SMD simulations.

c-d: Structure comparisons of Z839878730-KK-LC1-Laminin G triple system during the 1 ns SMD simulations. Cyan cartoon structure represents FAT1 conformations. Green, orange, yellow, magenta, raspberry and blue cartoon structure represent KK-LC-1 for the conformation at 0 ps, 200 ps, 400ps, 600 ps, 800ps and 1000ps.

e: Time dependence of rupture forces between the KK-LC-1 and FAT1 during SMD simulations.

REFERENCE:

3. Liu Y, Zhao L, Li W, Zhao D, Song M, Yang Y. FIPSDock: a new molecular docking technique driven by fully informed swarm optimization algorithm. J Comput Chem 34, 67-75 (2013).

Comment 6: “***What are the advantages of using computational approach for the discovery of protein-protein interface inhibitors in this work? The authors should elaborate in the Discussion Section.***”

RESPONSE: Thank you for this critical comment. Per the referee’s suggestion, we have elaborated to discuss the advantage of using computational approach for the discovery of protein-protein inhibitors (PPI) in this work. Please see draft-track-change (line 442-450) or below for more details.

“Protein-protein interaction is regarded as the basis for many life processes and therefore protein-protein interaction is an important class of drug targets with a wide range of applications and market potential. However, to date, targeting protein-protein interaction is still a rather challenging task. The key point of drug design to target protein-protein interactions is to find key amino acid residues at the site of action⁴. Furthermore, capturing the dynamic conformation of protein-protein interactions for virtual screening of compound library is one prerequisite for the design of protein-protein inhibitors (PPI). In the present work,

we employed molecular dynamics simulations to optimize the complex structure of KK-LC-1 and FAT1. Moreover, key amino acid residues at the binding interface of KK-LC-1 and FAT1 were evaluated to define the pocket of compound virtual screening.”

REFERENCE:

4. Wanner J, Fry DC, Peng Z, Roberts J. Druggability assessment of protein-protein interfaces. Future medicinal chemistry 3, 2021-2038 (2011).

Comment 7: “**Some of the fonts in the figures are too small.**”

RESPONSE: This has been fixed. We have amended the fonts in the Figures to ensure that they are all clearly visible.

Comment 8: “**The band quality of KK-LC-1 in Figure 3f and Figure 3h should be improved.**”

RESPONSE: This has been fixed. We have replaced the WB band of KK-LC-1 in Figure 3f and Figure 3h as below.

Figure. 3f

Figure. 3h

Comment 9: “*The molecular weights of the proteins for WB should be labeled in the figures.*”

RESPONSE: This has been fixed. We have labeled the molecular weights for the WB bands in all the figures.

Comment 10: “*The author should indicate the exact number of mice used in the in vivo drug susceptibility in the figures and the figure legends.*”

RESPONSE: This has been fixed. We have labeled the exact number of the mice used in all the in vivo experiments.

CLOSING RESPONSE TO REFEREE #2:

Again, we truly appreciate all of your very insightful comments. Thank you for taking the precious time and kind efforts to help us improve our work.

Reviewer #3 - Computational screening - (Remarks to the Author):

General Comments: “.....I ***think this is a quite sound work, with a robust experimental approach with multiple complementary experimental techniques that seem to fit and solidly demonstrate the validity of the proposed target for anticancer drug design. However, I still see some points that would require improvement before being ready for publication...***”

RESPONSE: We truly appreciate for your very kind words about our work. We are excited to hear that you think our manuscript is relatively novel with translational impact to anticancer drug design. We are grateful for your valuable time and kind efforts you have spent to provide your very insightful comments, which are of great help for our revision.

Comment 1: “***The description of the computational approach is really poor. Since both the KK-LC-1 and FAT1/Laminin G domain are both modeled (there are no experimental structures for them), how was the protein-protein interface identified?***”

RESPONSE: We appreciate for this critical and very helpful comment. This has been fixed. Per the referee’s comment, we have added more details to carefully describe the computational discovery of small-molecule inhibitor that may disrupt the KK-LC-1 and FAT1 binding interface. Please see draft-track-change (line 771-794) or below for more details.

“The KKLC1 protein structure was obtained from AlphaFold Protein Structure Database (<https://alphafold.ebi.ac.uk/>) with accession ID AF-Q5H943-F1. The query sequence of FAT1 was retrieved from UniProtKB with accession ID Q14517 and the extracellular Laminin G domain (aa: 3829-4009) was built by Modeller 10.1(SI Fig. 1), with several crystal structures from Laminin G protein family as the references (PDB ID: 1C4R, 1F5F, 2JD4, 3SH4, 5CM9). The protein-protein docking was conducted by ZDOCK 3.0.2 followed by MD

optimization. The virtual screening of compound library was conducted with AutoDock Vina, FIPSDock tools³. The MM/GBSA free energy and Interfaces of complex were analyzed by HawkDock Server (<http://cadd.zju.edu.cn/hawkdock/>) and PDBePISA tools (https://www.ebi.ac.uk/msd-srv/prot_int/). The 2D protein-ligand and protein-protein interaction profiles were generated using LigPlot+ and VMD educational version. Molecular dynamics (MD) simulations were performed with Gromacs 2020.3 package. All molecular graphics were displayed and prepared using the PyMOL educational version.

We first selected a number of compound libraries for virtual screening from the enamine kinase library (HY-L0061V) and the protein-protein interaction library (HY-L0066V), which consist of ~70,000 small molecules. Molecular dynamics simulations (MD) were performed to optimize the complex structure of KK-LC-1 and FAT1. The key amino acid residues at the complex binding interface were evaluated to define the virtual screening pocket ($center_x = 36.294$; $center_y = 49.379$; $center_z = 37.191$). The compounds were then evaluated and ranked according to binding free energies. The top 10% of small molecules were filtered by druggability properties to remove molecules with potential toxicity or poor drug-like abilities. Finally, 145 compounds were evaluated via the bio-layer interferometry (BLI) experimental assay for binding affinity. Among them, Z839878730 (Z8) was found to have the strongest binding ability and was predicted to disrupt the binding KK-LC-1 to FAT1 (Fig. 5 d-f and SI Fig. 2)."

SI Fig. 1. The modeling structure of FAT1 Laminin G domain. The double layer structure consists of 14 beta-sheet structures which are numbered from 1' to 12'. The upper layer in this figure is made by Sheet-12, Sheet-1, Sheet-10, Sheet-3, Sheet-4, Sheet-5, and Sheet-6, that the lower one is made by Sheet-9, Sheet-8, Sheet-7, Sheet-2, Sheet-1', and Sheet-12'.

SI Fig. 2. The key amino acid residues of KK-LC-1-FAT1 binding interface and screening pocket.

“After 100 ns MD simulations, we found that the interaction between KK-LC-1 and Laminin G domain was significantly disrupted in the complex system (Fig. 5 d-f). The small molecule occupied the groove composed by Sheet-1, Sheet-12 and several loop structures. Hence, the C-terminal part of KK-LC1 helix structure become difficult to interact with Laminin G domain, which contributes the most favorable interaction between these two proteins. The helical structure was even breakdown at the latter part of the trajectory in the complex system during the 100ns MD simulation.

In particular, to assess the putative capability of small-molecule inhibitor disrupting the KK-LC-1 and FAT1 binding interface, we further performed the Steered Molecular Dynamics (SMD) simulations on the equilibrated model after the 100ns MD simulations. A constant pulling velocity was applied to KK-LC-1 to pull it away from the binding surface of the Laminin G domain. To achieve this, 50 kJ mol⁻¹ nm⁻² soft elastic spring and 0.001 nm ns⁻¹ rupture force were

applied to KK-LC-1 along the pulling direction, whereas the Laminin G domain was chosen as the static reference throughout the pulling process. Snapshots of the structure were captured every 200 ps and steering forces were recorded every 50 ps. The models were pulled for 1 ns. During the 1 ns SMD simulations (Supplementary Fig. 8), we found that KK-LC-1 can easily depart from FAT1 in Z839878730-KK-LC1-Laminin G complex with lower force as compared to KK-LC1-FAT1 complex. The distances between the center of mass of the KK-LC-1 and that of FAT1 before and after pulling were 1.953 nm and 2.953 nm in KK-LC1-FAT1 complex, and were 1.259 nm and 2.259 nm in Z839878730-KK-LC1-Laminin G complex system, respectively. The pulling direction for KK-LC-1 was chosen to achieve effective separation and all of the interactions between the KK-LC-1 and FAT1 disappeared around 400ps. These results implicated that Z8 may be able to disrupt the binding between KK-LC-1 to FAT1.”

Supplementary Figure 8.

a-b: Structure comparisons of KK-LC1-FAT1 system during the 1 ns SMD simulations.

c-d: Structure comparisons of Z839878730-KK-LC1-Laminin G triple system during the 1 ns SMD simulations. Cyan cartoon structure represents FAT1 conformations. Green, orange, yellow, magenta, raspberry and blue cartoon structure represent KK-LC-1 for the conformation at 0 ps, 200 ps, 400ps, 600 ps, 800ps and 1000ps.

e: Time dependence of rupture forces between the KK-LC-1 and FAT1 during SMD simulations.

REFERENCE:

3. Liu Y, Zhao L, Li W, Zhao D, Song M, Yang Y. FIPSDock: a new molecular docking technique driven by fully informed swarm optimization algorithm. J Comput Chem 34, 67-75 (2013).

Comment 2: “***How was the binding pocket used for the virtual screening of compounds selected? Did it involve only KK-LC-1 or both KK-LC-1/FAT1?***”

RESPONSE: We are grateful for this critical comment. This has been fixed. We have carefully described the identification of binding pocket for virtual screening. Please see draft-track-change (line 789-794) or below for more details.

“The key amino acid residues at their binding interface of KK-LC-1 and FAT1 were evaluated to define the screening pocket after the 100ns MD simulations (center_x = 36.294; center_y = 49.379; center_z = 37.191). The binding pocket involved both amino acid residues of KK-LC-1 and FAT1 (SI Fig. 3 and SI Table 1).”

SI Fig. 3. The key amino acid residues of KK-LC-1-FAT1 binding interface. The green dash lines represent putative hydrogen binding. The arcs represent the putative hydrophobic interactions.

SI Table 1. The key amino acid residues of KK-LC-1-FAT1 binding interface that define the pocket of virtual screening for the discovery of PPI.

No.	KK-LC-1	FAT1	Dist. [Å]	No.	KK-LC-1	FAT1	Dist. [Å]
1	ILE-66	SER-66	1.7	8	ASN-83	ASN-117	3.82
2	ILE-66	GLY-67	3.27	9	ASN-83	LEU-118	2.93
3	PHE-70	SER-66	2.92	10	ASN-89	ARG-134	2.38
4	ARG-76	ASP-63	2.88	11	LYS-90	ASP-119	1.72
5	ARG-76	CYS-64	3.37	12	GLU-93	ARG-137	2.78
6	ARG-79	THR-46	2.94	13	THR-97	TYR-14	2.85
7	ARG-79	ASP-47	2.74	14	LYS-101	HIS-170	3.11

Comment 3: “***What compound collection was used? The description of the methods used seem rather a black box by just mentioning the computational tools used.***”

RESPONSE: We want to thank the referee for this expert comment and suggestion. Indeed, we selected a library of compounds targeting the KK-LC-1-FAT1 binding interface from the enamine kinase library (HY-L0061V) and the protein-protein interaction library (HY-L0066V), which in total contain ~70,000 small molecules. Moreover, per the referee’s comment, we have added more details to carefully describe the computational discovery of small-molecule inhibitor that may disrupt the KK-LC-1 and FAT1 binding interface. In particular,

to assess the putative capability of small-molecule inhibitor disrupting the KK-LC-1 and FAT1 binding interface (Supplementary Figure 8), we further performed the Steered Molecular Dynamics (SMD) simulations on the equilibrated model after the 100ns MD simulations. Please see draft-track-change for more details.

Comment 4: “***In Figure 1e, ALDH- cells show near 10K-fold higher expression of KK-LC-1 than ALDH+ cells, while in the text (eg. line 138) the opposite message is conveyed (ALDH+ cells have higher KK-LC-1 expression). Is this a bug in the Figure labels?***”

RESPONSE: Many thanks for this critical comment. This has been fixed. It should be ALDH+ cells showed a 10K-fold higher expression of KK-LC-1 than ALDH- cells. We hope that this concern has been addressed. Please see below for more details.

Figure 1e

Comment 5: “***Please provide explanation for "CTA" in lines 82 and 283***”.

RESPONSE: Thank you for this valuable suggestion. This has been fixed. “CTA” was the abbreviation for ‘cancer testis antigen’, a family of protein which specifically expressed in only testis and tumor tissue⁵. We have explained the term carefully in the revised manuscript.

REFERENCE:

5. Gibbs ZA, Whitehurst AW. Emerging Contributions of Cancer/Testis

Antigens to Neoplastic Behaviors. Trends Cancer 4, 701-712 (2018).

Minor issues:

Comment 6: "***multiple typos in the text (eg "resisitance", "resisitant" instead of resistance and resistant in page 5 multiple times, "popuplations" instead populations in the same page, etc)***"

RESPONSE: Many thanks for this critical comment. We have carefully checked and fixed all the typos throughout the manuscript. Thanks again for the valuable and helpful suggestion.

CLOSING RESPONSE TO REFEREE #3:

We are grateful for all of your insightful and expert comments. Again, many thanks for taking the precious time and kind efforts to help us improve our work.

Reviewer #4 - Breast cancer therapy (Remarks to the Author):

General Comments: “*The authors of this manuscript describe a novel signalling axis in TNBC that promotes breast cancer stemness and is associated with therapy resistance.Through their findings, the authors have identified a novel therapeutically relevant pathway that is validated in patient-derived tissues and identified a promising therapeutic molecule, Z8, targeting this pathway.*”

RESPONSE: Many thanks for your kind words about our work. We are excited to learn that you think our manuscript is relatively novel with translational impact. We are grateful for your valuable time and kind efforts you have spent to provide your expert comments, which are of great help for our revision.

Comment 1: “*In several instances, the authors include identical data in both the main figures and the extended figures, duplicates should be removed to avoid confusion.*”

RESPONSE: Many thanks for this critical comment. This has been fixed. We have removed duplicates in the figures to avoid confusion (Please see **Supplementary Figure 5b** below for details).

Supplementary Figure 5b

Comment 2: ***“The authors should indicate the sample sizes for animal experiments and tumour samples in the figure legends and methodology in cases where it is not stated.”***

RESPONSE: Many thanks for this helpful suggestion. We have indicated the sample sizes for animal experiments and tumor samples wherever applicable.

Comment 3: ***“The authors appear to interchange between using HSP90 and GAPDH as loading controls/house-keeping proteins, they should explain why the loading control is not kept consistent throughout the manuscript.”***

RESPONSE: We want to thank the referee for this critical comment. Indeed, in the scenario that we want to detect proteins with low molecular weights, we used GAPDH (about 35 Kda) in the experiments. On the other hand, in the scenario that we want to detect proteins with high molecular weight, we used HSP90 (about 90 Kda) in the experiments. We hope these could clarify the concerns.

Comment 4: ***“A positive control hippo pathway activator is an important control that could have been included throughout the manuscript and the authors should address why a positive control was not included.”***

RESPONSE: Many thanks for this critical and helpful comment. We fully agree with the referee that a positive Hippo pathway activator could be an important control. In fact, some studies involving Hippo signaling pathway didn't use Hippo pathway activator as control^{6, 7}. Nevertheless, per the referee's suggestion, we conducted extra Western blot experiments on Hippo signaling pathway using MY-8758, an Hippo pathway activator, as the positive control. We found that 5µM of Z8 exerted similar functions in activating the Hippo signaling pathway (MST1, pMST1, LATS1, pLATS1, YAP1 and pYAP1) as 100nM MY-875 did. The WB results are shown below in **Supplemental Figure 1**. Please see draft-track-change for more details.

Supplemental Figure 1. Western blot analysis of Hippo signaling pathway in MDA-MB-231 and MDA-MB -468 cells after treated with DMSO, Z8(5 μ M) or MY-875 (100 nM).

REFERENCE:

6. Li Z, et al. Loss of the FAT1 Tumor Suppressor Promotes Resistance to CDK4/6 Inhibitors via the Hippo Pathway. *Cancer Cell* 34, 893-905 e898 (2018).
7. Ma S, et al. Hippo signalling maintains ER expression and ER(+) breast cancer growth. *Nature* 591, E1-E10 (2021).
8. Song J, et al. Discovery of N-benzylarylamide derivatives as novel tubulin polymerization inhibitors capable of activating the Hippo pathway. *Eur J Med Chem* 240, 114583 (2022).

Comment 5: “***Nature Communications endorses the recommendations of the ARRIVE guidelines for animal experiments and therefore the authors should include more detailed descriptions of the animal husbandry and***

housing, randomisation and blinding, sample sizes, and rationale for strains chosen etc when describing their animal experiments. Importantly, the animal ethics authorization reference numbers should be quoted which are missing from this manuscript.”

RESPONSE: Many thanks for this critical comment. We conducted all the animal studies according to the ARRIVE guidelines and described the animal experiments according to it wherever applicable. Per the referee’s comment, we have added the animal ethics authorization reference numbers in the revised manuscript. Please see draft-track-change for more details.

Comment 6: ***“There are some spelling and grammatical errors throughout the manuscript, particularly on page 5 and in the discussion section, that if fixed would improve the manuscript's readability.”***

RESPONSE: Thanks for this critical comment. Per the referee’s comment, we have checked and fixed the spelling & grammatical errors throughout the manuscript. We hope these could improve the readability of our manuscript. Thanks again for the kind efforts.

Major comments

Comment 7: ***“page 5, lines 138-140: The authors describe KK-LC-1 to be overexpressed in the ALDH+ population, however, the graph in figure 1e shows the exact opposite result. This is likely a simple error but needs to be addressed. It should also be described/mentioned in the results or figure legend, how this data was generated.”***

RESPONSE: We are grateful for this helpful comment. This has been fixed. Per the referee’s comment, we have amended the mislabelled figure and elaborated how this data was generated in the Results section. Please see draft-track-change or below for more details.

Figure 1e

Comment 8: “***On page 5, lines 143-145, the authors describe KK-LC-1 expression as highly tumor specific. Do the authors mean specific to TNBC tumours or is this a pan-cancer phenomenon? This is not clear. If it is the latter then this contradicts the data in figure 1B and needs explaining.***”

RESPONSE: Many thanks for this critical comment and sorry for the confusion. This has been fixed. Herein, we want to claim that KK-LC-1 was specifically and highly expressed in the TNBC tumor. We have amended the phrase to clarify the concern in the revised manuscript.

Comment 9: “***From figure 1F onwards, why were the KK-LC-1 knockdown experiments not carried out in the docetaxel-resistant cell line, given that the data in 1a indicates KK-LC-1 overexpression to be specific to these cells compared to parental cells?***”

RESPONSE: We are grateful for this critical and expert comment. We fully agree with the referee that KK-LC-1 knockdown experiments are necessary. Per the referee’s suggestion, we have conducted extra experiments and we knocked down KK-LC-1 in docetaxel-resistant MDA-MB-231 cells. We found that knockdown of KK-LC-1 could reduce the docetaxel resistance in docetaxel-resistant MDA-MB-231 cells (**Supplementary Figure 2e**). Furthermore, we detected the ratio of ALDH⁺ cells using flow cytometry. Our results revealed that KK-LC-1 has a significant impact on the ALDH⁺ cell ratio in docetaxel-resistant MDA-MB-231 cells (**Supplementary Figure 2f-g**).

Supplementary Figure 2e

Supplementary Figure 2f-g

Comment 10: ***“The mass-spectrometry description for figure 3b is missing from the materials and methods and needs to be included with a description of the machinery used and the sample preparation steps.”***

RESPONSE: Thanks for this critical comment. This has been fixed. We have added the detailed methodology about the machinery used and the sample preparation steps in Materials and Methods part and Related Manuscript Files. Please see draft-track-change (line 685-694) or below for more details.

“Protein Mass Spectrometry

To detect potential proteins which may interact with KK-LC-1, we first performed the immunoprecipitation assay in MDA-MB-231 cells with anti-KK-LC-1 antibody and anti-IgG antibody according to manufactures' instructions. When the electrophoresis process was over, the gels were cut down according to the lanes containing different samples (anti-KK-LC-1 and anti-IgG) and all molecular weights of the gels were included. Then the gels were analyzed using mass spectrometry with the help of Beijing Protein Innovation Co., Ltd, the

proteins which may also immunoprecipitated with anti-IgG antibody was removed during the analysis and a report of potential proteins which may interact with KK-LC-1 was generated.”

Comment 11: “**The comparisons for figure 3h are made across two separate gels, ideally, the protein should have been run on one gel. Were these gels processed in parallel? This information should be disclosed in the manuscript.**”

RESPONSE: Thanks for this valuable comment. Indeed, the comparison of the same protein in Figure 3h were run on one gel. We have separated the same protein in the figure to clarify the results. Moreover, we have provided the Raw gel data in the Source Data. For different bands of KK-LC-1 and FAT1, we have run them in parallel gels due to their significant differences in molecular weights (13 kDa and 506 kDa). Per the referee’s suggestion, we have disclosed this in the Materials and Methods section. We hope these could clarify the concerns.

Comment 11: “**the conclusion drawn from the data in figures 4e-f is weak and cannot be substantiated without proper quantification of the immunocytochemistry results.**”

RESPONSE: We are grateful for this insightful comment. Per the referee’s comment, we have made proper quantification of the immunofluorescence results in Figure 4c-d in the revised manuscript as below.

Figure 4c-d

Minor comments

Comment 12: ***“In the introduction “CTA” should be written in full before use of the acronym.”***

RESPONSE: This has been fixed. “CTA” was the abbreviation for ‘cancer testis antigen’. We have amended this in the revised manuscript.

Comment 13: ***“On page 5, line 126, the authors mention “upregulated genes in TNBC.” The database used to acquire this list of genes needs to be mentioned somewhere.”***

RESPONSE: Many thanks for this comment. This has been fixed. The data used to analyze “upregulated genes in TNBC” were acquired from TCGA database and we have mentioned this in the Materials and Methods section.

Comment 14: ***“The methods for figure 2d do not seem to be described in the materials and methods in terms of how spheroids were processed for immunofluorescence.”***

RESPONSE: This has been fixed. We have described the methodology in the Materials and Methods section. Please see draft-track-change for more details.

Comment 15: ***“A quantification for figure 2e would be a useful addition with statistical significance displayed.”***

RESPONSE: Many thanks for this insightful comment. Per the referee’s suggestion, we have conducted the statistical analysis of the results in Figure 2e in Supplementary Table 2 as below.

Supplementary Table 2

Statistical results of xenograft assay using ALDH⁺ MDA-MB-231 cells upon *KK-LC-1* knockdown.

Cell number	sh-NC	sh- KK-LC-1
2×10 ⁴ cells	6/6	3/6
5×10 ³ cells	5/6	2/6
2×10 ³ cells	4/6	1/6
500 cells	4/6	0/6

Tumor initiating frequency (TIC)	1/1426 (1/733-1/2774)	1/20878 (1/9015-1/48354)
P-value		$P < 0.001$

Comment 16: “*As per Nature Communication guidelines, uncropped, unprocessed gels should be made available and this will be relevant for figure 3h FAT1 blots where there appear to be additional higher weight bands cropped out of the image that might be relevant to reviewers and the eventual readers of this manuscript.*”

RESPONSE: Many thanks for this critical comment. Per the referee’s suggestion, we have provided all the uncropped, unprocessed gels in Source Data wherever applicable. Indeed, there were no additional higher weight bands cropped out of the image in Figure 3h. These were merely the background of the lane in the gel instead of a band. We hope these could clarify the concern.

Figure 3h

CLOSING RESPONSE TO REFEREE #4:

Again, we appreciate all of your very insightful and helpful comments. Thank you for taking the precious time and kind efforts to help us improve our work.

Reviewers' Comments:

Reviewer #1:

Remarks to the Author:

The authors have addressed all my concerns on this manuscript. Therefore, it is ready to be accepted.

Reviewer #2:

Remarks to the Author:

The authors answered all of my concerns very well.

Reviewer #3:

Remarks to the Author:

The authors have addressed my concerns, and particularly have provided detailed description of the computational procedures, and therefore I think the paper should be published

Reviewer #4:

Remarks to the Author:

In their revised manuscript, the authors have addressed the concerns raised in my initial feedback. I am fully satisfied with the changes that have been made which include additional experiments, extensions to the methods sections, more detailed reporting, and corrections to figures. The manuscript is novel, highly relevant and will make an important, therapeutically relevant contribution to our understanding of the molecular drivers of drug-resistant TNBC.